**Particulate trace metal dynamics in response to increased $CO_2$ and iron availability in a coastal mesocosm experiment**

M. Rosario Lorenzo[1], María Segovia[1], Jay T. Cullen[2], and María T. Maldonado[3]

[1]Department of Ecology, Faculty of Sciences, University of Málaga, Bulevar Louis Pasteur s/n, 29071-Málaga, Spain

[2]School of Earth and Ocean Sciences, University of Victoria, 3800 Finnerty Road, A405, Victoria BC V8P 5C2 Canada

[3]Department of Earth, Ocean and Atmospheric Sciences, University of British Columbia, 2207 Main Mall, Vancouver BC V6T 1Z4, Canada

*Correspondence to:* María Segovia (segovia@uma.es) and María T. Maldonado (mmaldonado@eoas.ubc.ca)

**Abstract.** Rising concentrations of atmospheric carbon dioxide are causing ocean acidification and will influence marine processes and trace metal biogeochemistry. In June 2012, in Raunefjord (Bergen, Norway) we performed a mesocosm experiment, comprised of a fully factorial design of ambient and elevated $p$CO$_2$ and/or an addition of the siderophore desferrioxamine B (DFB). In addition, the macronutrient concentrations were manipulated to enhance a bloom of the coccolithophore *Emiliania huxleyi*. We report here the changes in particulate trace metal concentrations during this experiment. Our results show that particulate Ti and Fe were dominated by lithogenic material while particulate Cu, Co, Mn, Zn, Mo and Cd had a strong biogenic component. Furthermore, significant correlations were found between particulate concentrations (mol L$^{-1}$) of Cu, Co, Zn, Cd, Mn, Mo, and P in seawater and phytoplankton biomass ($\mu$gC L$^{-1}$), supporting a significant influence of the bloom in the distribution of these particulate elements. The concentrations of these biogenic metals (mol L$^{-1}$) in the *E. huxleyi* bloom were ranked as: Zn > Cu $\approx$ Mn > Mo > Co > Cd. Changes in $CO_2$ affected total particulate concentrations (mol L$^{-1}$) and biogenic metal ratios (Me:P) for some metals, while the addition of DFB only affected significantly the concentrations of some particulate metals (mol L$^{-1}$).Variations in $CO_2$ had the most clear, and significant effect on particulate Fe concentrations (mol L$^{-1}$), decreasing its concentration under high $CO_2$. Indeed, high $CO_2$ and/or DFB promoted the dissolution of particulate Fe, and the presence of this siderophore helped maintaining high dissolved Fe. This shift between particulate and dissolved Fe concentrations, in the presence of DFB, promoted a massive bloom of *E. huxleyi* in the treatments with ambient $CO_2$. Furthermore, high $CO_2$ decreased the Me:P ratios of Co, Zn and Mn, while increased the Cu:P ratios. These findings support theoretical predictions that the Me:P ratios of metals whose seawater dissolved speciation is dominated by free ions (e.g. Co, Zn and Mn) will likely decrease or stay constant under ocean acidification. In contrast, high $CO_2$ is predicated to shift the speciation of dissolved metals associated with carbonates, such as Cu, increasing their bioavailability, and resulting in higher Me:P ratios.

*Key words:* Global change, Fe, $CO_2$, particulate trace metals, dissolved trace metals, mesocosms, *Emiliania huxleyi,* phytoplankton

## 1. Introduction

Marine phytoplankton contribute half of the world's total primary productivity, sustaining marine food webs and driving the biogeochemical cycles of carbon and nutrients (Field et al., 1998). Annually, phytoplankton incorporate approximately 45 to 50 billion metric tons of inorganic carbon (Field et al., 1998), removing a quarter of the $CO_2$ emitted to the atmosphere by anthropogenic activities (Canadell et al., 2007). Yet, the atmospheric $CO_2$ concentration has increased by 40 % since pre-industrial times as a result of anthropogenic $CO_2$ emissions, producing rapid changes in the global climate system (Stocker et al., 2013). The dissolution of anthropogenic $CO_2$ in seawater, causes shifts in the carbonate chemical speciation, and leads to ocean acidification (OA). Marine ecosystems are sensitive to changes in pH because pH strongly affects chemical and physiological reactions (Hoffman et al., 2012). Increased $CO_2$ in seawater may enhance or diminish phytoplankton productivity (Mackey et al., 2015), decrease the $CaCO_3$ production in most planktonic calcifiers (Riebesell and Tortell 2011), and/or inhibit organic nitrogen and phosphorus acquisition (Hutchins et al., 2009). Thus, the biogeochemical cycling of nutrients is predicted to be highly affected by OA (Hutchins et al. 2009), as well as the distribution and speciation of trace metals in the ocean (Millero et al., 2009).

Trace metals, including Fe, Zn, Mn, Cu, Co and Mo, are essential for biological functions (e.g. photosynthesis, respiration and macronutrient assimilation), and Cd can supplement these functions. Trace metals availability can influence phytoplankton growth and community structure (Morel and Price, 2003). In turn, plankton control the distribution, chemical speciation, and cycling of trace metals in the sea (Sunda, 2012), by, for example, releasing organic compounds that dominate the coordination chemistry of metals, internalizing trace elements into the cells, and reducing and/or oxidizing metals at the cell surface. The chemistry of redox speciation of active trace metals is highly dependent on pH.     Fe occurs in two main redox states in the environment: oxidized ferric Fe (Fe (III)), which is poorly soluble at circumneutral pH; and reduced ferrous Fe (Fe (II)), which is     more soluble in natural seawater, but becomes rapidly oxidized (Millero et al. 1987). Fe speciation and bio- availability are dynamically controlled by the prevalent changing redox conditions. Also, as the ocean becomes more acidic, reduction of Cu (II) will increase, as the ionic form of Cu (II) is reduced to Cu (I) (Millero et al., 2009). The effect of higher concentrations of Cu (I) in surface waters on biological systems is not well known. Therefore, while the effects of OA on inorganic metal speciation will be more pronounced for metals that form strong complexes with carbonates (e.g. copper) or hydroxides (e.g. Fe and aluminium), those that form stable complexes with chlorides (e.g. cadmium) will not be greatly affected. pH mediated changes in concentrations and/or speciation could possibly enhance trace metals limitation and/or toxicity to marine plankton (Millero et al., 2009).

Fe is crucial for phytoplankton growth because     it is involved     in many essential physiological processes, such as photosynthesis, respiration, and nitrate assimilation (Behrenfeld and Milligan, 2013). The decrease in seawater pH in response to OA may increase Fe solubility (Millero et al., 2009), but it may also result in unchanged or lower Fe bioavailability, depending on     the nature of the strong organic Fe ligands (Shi et al., 2010). Consequently, changes in Fe bioavailability due to ocean acidification can affect positively or negatively ocean productivity and $CO_2$ drawdown. Copper is an essential micronutrient but may be toxic at high concentrations (Semeniuk et al., 2016). An increase in free cupric ion concentrations in coastal areas due to ocean acidification (Millero et al., 2009) could result in negative effects on phytoplankton. Given that  trace metals are essential for phytoplankton productivity, and that they are actively internalized during growth, it is important to study the impacts of ocean acidification on the trace metal content of ecologically significant plankton species.

In a rapidly changing global environment, generated by anthropogenic $CO_2$ emissions, it is critical to gain adequate

understanding about ecosystem responses. Due to the complex interactions in aquatic ecosystems, such predictions have
so far not been possible to do based upon observational data and modelling alone. However, direct empirical studies on
natural communities offer a robust tool to analyse interactive effects of multiple stressors. Specifically, mesocosm
experiments allow perturbation studies with a high degree of realism compared to other experimental systems such as in
the laboratory (high controlled conditions usually far from reality) or *in situ* in the ocean (where not all the interactions
are contemplated) (Riebesell et al., 2010, Stewart et al., 2013, Riebesell and Gatusso, 2015).

In the present work a bloom of the coccolithophorid *Emiliania huxleyi* was induced in a mesocosm experiment in a
Norwegian fjord, where the speciation of particulate and dissolved trace metals is very dynamic (e.g. Fe; Ozturk et al.
2002). We aimed to examine and characterize the change of particle trace metals during an E. huxleyi bloom under the
interactive effects of increased $CO_2$ and/or dissolved Fe. *Emiliania huxleyi* is the most cosmopolitan and abundant
coccolithophore in the modern ocean (Paasche, 2002) and its growth and physiology has been studied under this
experimental conditions (Segovia et al., 2017, Segovia et al., 2018, Lorenzo et al., 2018). Furthermore, *E. huxleyi* has
unique trace metal requirements relative to other abundant phytoplankton taxa (ie. diatoms or dinoflagellates; Ho et al.
2003).    Coccolithophores play a key role in the global carbon cycle because they produce photosynthetically organic
carbon, as well as particulate inorganic carbon through calcification. These two processes foster the sinking of
particulate organic carbon—and trace metals—and contribute to    deep ocean carbon export (Hutchings, 2011) and
ultimately to    organic carbon burial in marine sediments (Archer, 1991, Archer and Maier-Reimer., 1994). However,
ocean acidification will disproportionally affect the abundance of coccolithophores, as well as their rates of
calcification and organic carbon fixation (Zondervan et al., 2007). The aim of the present study was to characterize the
changes in particulate trace metal concentrations—in both lithogenic and biogenic particles—    during a    bloom of
*E. huxleyi*    under realistic changes in $CO_2$ and Fe bioavailability expected by 2100.

**2. Materials and methods**
**2.1 Experimental set-up**
The experimental work was carried out in June 2012 in the Raunefjord, off Bergen, Norway as described in detail by
Segovia et al., (2017). Twelve mesocosms (11 m$^3$ each) were set-up in a fully factorial design with all combinations of
ambient and elevated $p$CO$_2$ and dFe in three independent replicate mesocosms. The mesocosms were covered by lids
(both transparent to PAR and UVR) and filled with fjord water from 8 m depth. We achieved two $CO_2$ levels
corresponding to present (390 ppm, LC) and those predicted for 2100 (900 ppm, HC) by adding different quantities of
pure $CO_2$ gas (Shculz et al., 2009). The specific $CO_2$ concentration and the $CO_2$ inlet flows in the mesocosms were
measured by non-dispersive infrared analysis by using a Li-Cor (LI-820) $CO_2$ gas analyser (Li-COR, Nebraska, USA).
$CO_2$ concentrations in the mesocosms were calculated from pH and total alkalinity measurements using the $CO_2$ SYS
software (Robbins et al,. 2010). At the beginning of the experiment, nitrate (10 μM final concentration) and phosphate
(0.3 μM final concentration) were added to induce a bloom of the coccolithophore *Emiliania huxleyi*, according to Egge
& Heimdal (1994). Following recommendations by Marchetti and Maldonado (2016), t    o induce changes in Fe
availability, and analyse its effects on the plankton community, 70 nM (final concentration) of the siderophore
desferrioxamine B (DFB) (+DFB and −DFB treatments) (Figure S1b-supplemental material) was added to half of the
mesocosms on Day 7, when the community was already acclimated to high $CO_2$. The initial dFe concentration before
DFB addition was about 4.5 nM. Even though DFB is a strong Fe-binding organic ligand often used to induce Fe
limitation in phytoplankton (Wells 1999), DFB additions may also in- crease the dissolved Fe pool in environments
with high concentrations of colloidal and/or particulate Fe, such as fjords (Kuma et al. 1996, Öztürk et al. 2002). By day

17, dissolved Fe concentrations were significantly higher (by ~3-fold) in the high $CO_2$ and DFB treatments than in the control (Segovia et al. 2017). These results support an increase in the solubility of Fe in seawater by either lowering its pH (Millero 1998; Millero et al. 2009) and/or the addition of DFB (Chen et al. 2004). The multifactorial experimental design consisted of triplicate mesocosms per treatment and the combinations of high and ambient $pCO_2$ and dFe levels, resulted in a total of 12 mesocosms: LC-DFB (control), LC+DFB, HC+DFB and HC-DFB. Water samples from each mesocosm were taken from 2 m depth by gentle vacuum pumping of 25 L volume into acid-washed carboys by using membrane vacuum pumps (PALL) working at reverse flow. Carboys were quickly transported to the onshore laboratory. The biological and chemical variables analysed were phytoplankton abundance and species composition, dissolved Fe and Cu concentrations (dFe, dCu), nutrient concentrations (nitrate, phosphate, silicic acid and ammonium) and particulate trace metal concentrations.

## 2.2 Dissolved copper (dCu)

Low density polyethylene (LDPE) bottles were cleaned with 1% alkaline soap solution for one week, then filled with 6 M trace metal grade HCl (Seastar, Fisher Chemicals) and submerged in a 2 M HCl bath for one month. For transport, they were filled with 1 M trace metal grade HCl for one more month and kept double bagged. In between each acid treatment, the bottles were rinsed with Milli-Q water (Millipore; hereafter referred to as MQ). Before sampling, the bottles were rinsed three times with filtered seawater. Seawater was collected from each mesocosm, filtered through AcroPak® capsule filters with 0.2 µm Supor® membrane into the trace metal clean LDPE bottles, and acidified with ultra-clean trace metal grade HCl in a Class 100 laminar flow hood. Total dissolved Cu concentrations were measured following Zamzow et al., (1998) using a flow injection analysis chemiluminescence detection system (CL-FIA, Waterville Analytical). Total dissolved Fe concentrations were measured as described in Segovia et al., (2017) for this very experiment. The pH of the 0.2 µm filtered dFe samples was lowered to 1.7 by using SeaStar HCl upon collection. Lowering the pH to 1.7, with HCl for more than 24 hours, ensures solubilisation of all the Fe in the sample, as well as the release of all the Fe bound within strong organic complexes (such as Fe-DFB), thus making all dFe available for analysis (Johnson et al. 2007). During (FIA-CL, the sample is only buffered to a higher pH immediately before entering the flow cell, right in front of the photomultiplier; so that Fe-DFB complexing kinetics are sufficiently slow to allow total dFe to be measured.

## 2.3 Particulate metals (pMe)
### 2.3.1 Sampling

All equipment and sampling material used during this study was rigorously acid-washed under trace metal clean conditions and protocols according to GEOTRACES . The material was cleaned with Milli-Q water (MQw) with 10 % Extran (Fisher Chemicals) at 60°C for 6h, followed by 3 thorough rinses with MQw at room T. The material was then cleaned with 10 % HPLC grade HCl (Sigma-Aldrich) at 60°C or 12h and then rinsed thoroughly 5 times with MQw at room T. The material was then covered by plastic and transported to the raft. Sampling in the raft was carried out under a mobile plastic cover hood. Filters were precleaned with 10% trace metal grade hydrochloric acid (Seastar, Fisher Chemicals), at 60°C overnight and were rinsed with MQw. Seawater samples (1-3.5 L) were filtered gently onto 0.45 µm acid washed Supor ®-450 filters (within a trace metal clean Swinnex filter holder) on days 12, 17 and 21 of the experiment. Four technical replicates were taken from each mesocosm. Two filters were analysed without oxalate-EDTA wash and the other two were individually washed with oxalate-EDTA reagent to remove extracellular Fe, as well as other metals (Tang and Morel, 2006). Immediately following filtration, the treated filters were soaked with 20

mL EDTA–oxalate solution, added to the headspace of the Swinnex holders, with an acid-washed polypropylene
syringe. After 10 min, vacuum was applied to remove the oxalate solution and 10 mL of 0.2 μm filtered chelexed
synthetic oceanic water (SOW) solution was passed through the filter to rinse off any remaining oxalate solution.
Replicate filters that were not treated with oxalate solution were transferred directly to 2 mL centrifuge polypropylene
tubes for storage. The filters with particles were frozen in acid-washed 2 mL polypropylene tubes and then, dried and
stored until analysis.

**2.3.2 Analytical methods**
Filters were digested in 7-mL acid-washed Teflon (Teflon, Rochester, NY, USA) vials. Teflon vials were also
precleaned using 10% trace metal hydrochloric acid (Fisher, trace metal grade) during two days and then, with nitric
acid (Fisher, trace metal grade) at 70 ºC during three days. In between each acid treatment, the bottles were rinsed with
MQ. Samples were digested in 3 mL of $HNO_3$ and 0.5 mL of HF (Fisher, trace metal grade) with lids on for 1 h on a
hot plate at 200 ºC. The lids were then removed to evaporate HF at 200ºC. After this, 1.5 mL of $HNO_3$ were added and
the samples were heated with lids on overnight at 150 ºC. Finally, 2.25 mL of $HClO_4$ (Fisher, Optima grade) were
added and the samples were heated for 4 h at 200 ºC. After complete digestion, the samples were dried on hot plates at
200ºC. The dried samples were dissolved in 1% nitric acid with 1 ppb in internal standard. The analysis was performed
using a high-resolution inductively coupled plasma-mass spectrometer (ICP-MS, Element XR, Thermo Scientific) and
the described instrumental settings (Table S1). Filter blanks were collected and subjected to the same storage, digestion,
dilution, and analysis processes, and these blank values were subtracted from sample measurements. Particulate
samples for ICP-MS analysis were processed in a trace metal-clean laboratory under a trace metal-clean laminar flow
fume hood.

**2.3.3 The effect of oxalate-EDTA wash on particulate trace metal concentrations**
To better estimate the biogenic fraction of the particulate metals, the filters were washed with an oxalate-EDTA
solution, which removes extracellular metals and oxyhydroxides (Tovar-Sanchez et al., 2003; Tang and Morel, 2006).
In our study, the oxalate wash significantly decreased the concentration of all particulate metals, with the exception of
Al and Ti (Tables S2 and S3), as observed by Rauschenberg and Twining (2015). The quantity of metal remaining after
the oxalate wash (i.e. biogenic fraction) varied among elements (Tables S2 and S3). In general, the concentrations of Fe
and Co in the particles were decreased the least by the oxalate wash by ~ 25%, while Mo and Pb concentrations were
decreased the most by ~70%. The concentrations of particulate Cu, Zn, Cd and Mn were reduced by 50% by the oxalate
wash. As shown previously (Sanudo-Wilhelmy et al. 2004), the oxalate reagent also removed extracellular P (by ~20%,
Table S2 & S3). Compared to Rauschenberg and Twining (2015), the estimates of the biogenic fraction, after the
oxalate wash, were in agreement for Co, Cu and P, and lower for Fe, Mn, Zn and Cd concentrations.

However, the efficacy of the oxalate wash to dissolve Fe, and other metals, from lithogenic particles is not well
constrained (Frew et al. 2006, Rauschenberg and Twining., 2015, King et al., 2012). Therefore, the results obtained
after the oxalate-EDTA wash should be interpreted with caution because we do not know whether the removed metal
fraction is a) only lithogenic; b) mainly lithogenic but some biogenic fraction is also removed, or c) whether metals
absorbed onto particles are equally labile to the wash on biogenic and lithogenic particles.  Given that many of the
trends we observed were identical for the oxalate-EDTA washed and non-washed particles [i.e. higher Me
concentrations in the LC+DFB treatments (Table S2 & S3) and positive correlations between phytoplankton biomass
and Me concentrations (Lorenzo-Garrido 2016)], below we present and discuss only the non-oxalate wash results.

**2.4 Statistical analyses**
Data were checked for normality (by Shapiro-Wilks' test), homoscedasticity (by Levene's test) and sphericity (by
Mauchly's test). All data met the requirements to perform parametric tests. Statistical significance of treatment effects
was carried out using Split-Plot ANOVA followed by post-hoc Sidak and Bonferroni tests (considering $P < 0.05$ as
significant). All analyses were performed using the General Linear Model (GLM) procedure. The correlation between
variables was analysed by Pearson's product-moment multiple comparisons (considering $P < 0.05$ as significant).
Statistical analyses were carried out using SPSS v22 (IBM statistics) and Sigmaplot 12 (Systat Software, Chicago,
USA).

**3. Results**
**3.1 Biological and chemical characteristics during the bloom**
Plankton community dynamics and their response to the applied treatments in the mesocosms are described in detail by
Segovia *et al.* (2017). Briefly, at the beginning of the experiment (days 1-10) a bloom of large chain-forming diatoms
was observed, which declined by day 7 (Figure S1g- supplemental material). This diatom bloom decline was
associated with a sharp decrease in nitrate and silicic acid concentrations (Figure S2-supplemental material, see Segovia
et al., 2017 for further details). Picoeukaryotes, dominated the phytoplankton community on day 8 (Figure S1d).
During the first 10 days of the experiment, there were no significant differences in the chemical variables measured
between the treatments (Figures S3 and S2 -supplemental material). On day 7, half of the mesocosms were amended by
adding DFB (+DFB treatments). Between day 7 and 17, an increase in dFe was observed in all treatments, except in the
control (Figure S3). This increase in dFe was sustained for the entire experiment in the DFB treatments (Figure S3).
Dissolved Cu concentrations were not affected by the different treatments (Figure S3). After day 10, a massive bloom
of the coccolithophore *E. huxleyi* developed under LC +DFB condition (Figure S1b), out-competing the rest of the
plankton groups (Figure S1). This bloom was not observed either in the control treatment (LC-DFB) or in the HC
treatments, although *E. huxleyi* was still the most abundant species in all treatments; with the exception of the HC-DFB
treatment (Figure S1b).

**3.2 Particulate metal concentrations during the mesocosm experiment**
The pMe concentrations (nM, mean of all treatments and dates) during the experiment were highest for Al, Fe and Zn,
and lowest for Cd, following this trend: Al ≈ Fe ≈ Zn > Ti > Cu ≈ Mn > Mo ≈ Pb > Co > Cd (Figure 1, Table S2).
Significant changes over time were observed for all particulate trace metal concentrations (Fe, Cu, Co, Zn, Cd, Mn, Mo
and Pb), except for Ti and Al (Figure 1, Table 1). The only metal that showed a significant time-dependent decrease in
its particulate concentration was Fe (Figure 1, Table 1). In general, the treatments with the highest particulate metals
concentrations also exhibited the highest particulate P, except for Al, Ti, Fe, and Pb (Figure 1, Table S2). On days 12
and 17, the highest particulate metals concentrations were observed in the LC+DFB treatment, while on day 21, they
were observed in both LC treatments (Figure 1, Table S2).

**3.3 The effects of increased CO$_2$ and the DFB addition on particulate metal concentrations**
Increased CO$_2$ and the DFB addition did not significantly affect the concentrations of particulate Al, Ti, Cu, and Pb
(Tables 1and S2). Similarly, the addition of DFB did not directly influence particulate concentrations of Fe, but high
CO$_2$ had a significant negative impact on particulate Fe (Tables 1 and S2, Figure 1). Particulate Cd concentrations were
also inversely affected by CO$_2$, but only in the presence of DFB (CO$_2$; and CO$_2$ x DFB effect, Tables 1 and S2, Figure
1). All other elements (P, Co, Zn, Mn and Mo) exhibited significant effects by CO$_2$ and by DFB, but there was also a
significant interaction between these two factors (Table 1, S2). This indicates that, for example, particulate Mn, Zn, Mo,
Co, and P concentrations were significantly decreased by high $CO_2$, but only in the +DFB treatments (Figure 1, Table 1,
S2,).  Similarly, the addition of DFB significantly increased pZn and pMn, but only at ambient $CO_2$ levels (Figure 1,
Tables 1, S2).
**3.4 Phosphorous-normalized metal ratios in particles collected from the mesocosms and the effects of increased**
**$CO_2$ and the DFB addition on these ratios**
The P-normalized metal ratios (Figure 2 and means in Table 2) were highest for Al and Fe (mean: $70 \pm 38$ mmol Al:
mol P, and $39 \pm 34$ mmol Fe: mol P), and lowest for Cd and Co (mean $0.02 \pm 0.01$ mmol Cd: mol P, and $0.07 \pm 0.02$
mmol Co: mol P). Fe:P and Ti:P were not significantly affected by increased $CO_2$ and/or the DFB addition, but showed
a significant decrease over time (Table 3). The P-normalized Cu, Co and Zn ratios changed significantly over time
(Table 3). Increased $CO_2$ significantly decreased Co, Zn and Mn:P ratios, while it increased Cu:P ratios (Figure 2, Table
3).  DFB did not affect the Me:P ratios of any of these bioactive elements (Table 3).
**4. Discussion**
**4.1 The effects of $CO_2$ and dFe on the plankton community**
In this experiment we investigated changes in particulate trace metal concentrations, in response to increased $CO_2$
and/or an addition of the siderophore DFB in a coastal mesocosm experiment. For a better understanding of the
processes affecting these stressors, we briefly summarise the mesocosm experiment results originating from Segovia et
al. (2017). High $CO_2$, as well as the DFB addition elevated dFe concentration increasing Fe availability (see Segovia et
al. 2027 for further details) . The higher dFe concentrations were sustained in the DFB treatments, suggesting that DFB
significantly increased the solubility of Fe, as previously shown (Chen et al. 2004). A bloom of the coccolithophore
*Emiliania huxleyi* was observed in the ambient $CO_2$ treatments, and was especially massive in the presence of DFB
(LC+DFB).      Our results suggest that *E. huxleyi* is able to utilise DFB-bound Fe (Fe-DFB). Indeed, *E. huxleyi* has
been shown to produce a wide range of organic compounds with high affinity for Fe (Boye & Van den Berg 2000).
Furthermore, *E. huxleyi* is able to acquire Fe from organic Fe complexes (Hartnett et al. 2012), including Fe-DFB
(Shaked & Lis 2012, Lis et al. 2015). Indeed, the biomass of *E. huxleyi* was negatively affected by increased $CO_2$.
However, increased dFe partially mitigated the negative effect of elevated $CO_2$, indicating that the coccolithophore was
able to acclimate better to ocean acidification when Fe availability was high. High dFe also had a positive effect on the
cyanobacterium *Synechococcus sp*, while the rest of the plankton food web did not response to the treatments (Segovia
et al. 2017).
**4.2 Particulate Fe and Ti are associated with lithogenic sources, while particulate Co, Cu, Zn, Cd, Mo and Mn**
**are associated with biogenic sources**
The particulate trace metal concentrations (nM, mean of all treatments and dates) during the experiment were highest
for Al, Fe and Zn, and lowest for Co and Cd, following this trend: Al $\approx$ Fe $\approx$ Zn > Ti > Cu $\approx$ Mn > Mo $\approx$ Pb > Co > Cd.
Lithogenic particles are enriched in Al and low in P (average crustal Al and P content is 2.9 mmol Al and 0.034 mmol
P g$^{-1}$ dry weight, Taylor 1964), while biogenic particles are enriched in P and low in Al (average plankton Al and P
content is 0.001 mmol Al and 0.26 mmol P g$^{-1}$ dry weight, Bruland et al. 1991). Therefore, the distinct high abundance
of Al and P in lithogenic and biogenic particles, respectively, can be used to evaluate the relative contribution of
lithogenic and biogenic material in our particulate samples.  In order to do this, first, it is important to establish that the
vast majority of the measured particulate P is associated the biogenic fraction. In this study, the abiotic P was estimated
using particulate Al concentrations (nM) and the P:Al ratio in crustal material, and was calculated to be negligible (<
1% of the total measured particulate P). In addition, a significant correlation ($p < 0.003$) was found between particulate
P concentrations and phytoplankton biomass (Table 4). Therefore, we assume a constant trace metal composition in
biogenic particles (assuming they are rich in phytoplankton) and lithogenic particles (assuming they are rich in crustal
material). We then calculated the expected metal concentrations in the particulate samples assuming that all the P
measured in the particles is associated with a biogenic fraction, and that all Al in the particles is associated with the
lithogenic fraction. Thus, for a given trace metal, its expected particulate trace metal concentration in seawater (mol L$^{-1}$)
can be calculated as the sum of the contribution from biogenic and lithogenic particles, so that:

$[Me] = a\,[P] + b\,[Al]$

where [Me] is the total concentration of the metal (mol L$^{-1}$) expected in the particulate sample; [P] is the P
concentration measured in the particles (mol L$^{-1}$); [Al] is the Al concentration measured in the particles (nM L$^{-1}$); $a$ is
the average, well-known metal content in biogenic particles, normalized to P (i.e. mol Me: mol P in marine plankton;
Ho 2006) and $b$ is the average, well known metal content in lithogenic particles, normalized to Al (mol Me: mol Al in
the Earth crust; Taylor 1964). For example, on day 21 in the HC-DFB treatment, the concentrations of particulate Al
and P were 8.22 and 134.8 nM, respectively (Table S2). Assuming a constant 0.0051 mol Fe: mol P in biogenic
particles (Ho 2006) and 0.331 mol Fe: mol Al in lithogenic particles (Taylor 1964; Table 2), we calculated an expected
particulate Fe concentration of 3.41 nM, where 80% was associated with lithogenic material and 20% with biogenic
material. Similar calculations were made for the bioactive metals Mn, Co, Cu, Zn, Cd, and Mo (Table 2). Our
calculations indicate that on average, particulate Fe was dominated by the lithogenic component (accounting for an
average of 78% of the total expected particulate Fe), while for particulate Co, Cu, Zn, Cd, and Mo the biogenic fraction
dominated (accounting for 94, 95, 99, 94 and 98%, respectively, of the total expected concentration; Table 2).
Particulate concentrations of Mn were also dominated by the biogenic fraction (65% of the total), but the lithogenic
fraction was also significant (35%). Moreover, the expected particulate Mn and Fe concentrations closely matched the
particulate Mn and Fe concentration we measured (accounting for an average of ~ 71% of the measured Mn, and 115%
of the measured Fe). For other metals (i.e. Cu, Mo and Zn), the expected particulate concentrations (nM) were lower
than measured (23% of the measured pCu, and 8% of measured pZn; Table 2). This suggests that the particles were
enriched in Cu, Mo, and Zn relative to what is expected based on natural marine plankton metal quotas (Bruland et al.
1991) and crustal ratios (Taylor 1964).

To further establish the lithogenic or biogenic source of the pMe in the particles, the particulate metal concentrations
were normalized to the concentrations of particulate P and Al (Figure 2, and Table 2). These ratios were then compared
with well-known molar ratios of metal to Al in the crust (Taylor 1964) and of metal to P ratios in marine plankton
samples (Ho 2006) and cultures (Ho et al. 2003) (Table 2). The average Fe: Al (506 mmol Fe: mol Al) and Ti:Al ratios
(119 mmol Ti: mol Al, Table 2) were relatively similar to crustal molar ratios (331 mmol Fe: mol Al and 39 mmol Ti:
mol Al; Taylor 1964). Additional evidence for the significant lithogenic component in particulate Ti and Fe was
gathered from Figure 3, where we plotted the molar ratios of the metals relative to P in the collected particles against
the Al:P ratios measured in those same particles. The slope of these data [(Fe:P)/(Al:P) = mol Me: mol Al] is the ratio
of Me:Al in the particles and can be compared to well-known Me:Al crustal ratio. Visually, if the data nicely fit the
Me:Al line for crustal material, these metals are mainly associated with the lithogenic component, as evident for Fe and
Ti (Figure 3). These combined results suggest that in our experiment, particulate Fe and Ti concentrations were
enriched by lithogenic material. In support of this finding, we also found no significant correlation between particulate
Fe and Ti concentrations (nM) and either the total plankton (phytoplankton and microzooplankton) or *E. huxleyi*
biomass (μg C L$^{-1}$; Table 4).

In contrast, when the P-normalized metal ratios in the particles collected from the mesoscosms were plotted against the
Al:P ratios in these particles, there were no correlations for the following metals Co, Cu, Zn, Cd, Mn and Mo (Figure
3), indicating that these particulate metals were not enriched in lithogenic material. Our measured metal: P ratios were
comparable to plankton ratios in natural samples and in cultures (Table 2). The concentrations (mol $L^{-1}$) of these metals
(i.e. Cu, Co, Zn, Cd, Mn, Mo), as well as P, also showed significant correlations with the biomass ($\mu$gC $L^{-1}$) of *E.*
*huxleyi* and that of total plankton cells ($p < 0.05$, Table 4), supporting a significant influence of the phytoplankton in the
distribution of these particulate elements.

**4.3 Particulate metals with a strong biogenic component: their P-normalized ratios**
The concentrations of particulate bioactive metals (mol $L^{-1}$), with a significant biogenic component (i.e. excluding Fe)
in the studied *E. huxleyi* bloom were ranked as: Zn > Cu ≈ Mn > Mo > Co > Cd (Figure 1, Table S3), similar to those
reported in indigenous phytoplankton populations: Fe ≈ Zn > Cu ≈ Mn ≫ Co ≈ Cd, (Twining and Baines, 2013). The
only treatment where E. *huxleyi* did not dominate the community was the HC-DFB; in this treatment the ranking of
these biogenic particulate trace metals was the same as that of LC+DFB (with the massive *E. huxleyi* bloom), but their
concentrations were lower than those in LC+DFB. At the end of the experiment, the concentrations of these biogenic
metals were, in general, comparable in both HC treatments, and lower than those in the LC treatments (Figure 1, Table
S3). Therefore, high $CO_2$ had a tendency to decrease particulate metal concentrations, especially on day 21. Given the
strong correlation between concentrations of these particulate bioactive metals and phytoplankton biomass, the lower
particulate concentrations in high $CO_2$ were mainly due to low phytoplankton biomass.

Particulate Zn concentrations were especially high in the LC+DFB treatment (Figure 1), where the highest *E. huxleyi*
biomass was observed. *Emiliania huxleyi* is well known for its high Zn cellular requirements (~ 1-10 for *E. huxleyi* vs.
1-4 mmol Zn: mol P for other phytoplankton; Sunda and Hunstman 1995, Sunda 2013). But, the Zn: P ratios in the
LC+DFB treatment (range 45-69 mmol Zn: mol P; Figure 2, Table S2), as well as in all the other treatment (range 16-
34 mmol Zn: mol P; Figure 2, Table S2) were significantly higher than these published ratios. This could be explained
by, the adsorption of these metals to the outside of the cells, and/or anthropogenic inputs of Zn into the fjord. The Zn:P
ratios in the samples washed with the oxalate-EDTA were still high (range 28-57 for LC+DFB and 16-33 mmol Zn:
mol P in all other treatments, Table S3), thus adsorption might have not been significant. We hypothesize that
anthropogenic aerosols which are rich in anthropogenic particulate metals, such as Zn and Cu (Perry et al. 1999; Narita
et al. 1999), and have high percentage of Zn and Cu dissolution (ref.), might be the source of these high Zn
concentrations and ratios in the particles.

Similarly, the Cu:P ratios in the collected particles were relatively elevated (1.4 ± 0.8 mmol Cu: mol P) compared to
those of other phytoplankton, including *E. huxleyi* (Table 2). The dissolved (7.7±0.41 nM Cu, Figure S3) and
particulate Cu concentrations (0.35±0.25 nM, Table S2) in our experiment were high, and similar to those previously
measured in this fjord (Muller et al., 2005). Rain events (or wet deposition of anthropogenic aerosols) in this fjord result
in high dissolved Cu and the active production of strong organic ligands by *Synechococcus*—to lower the free Cu
concentrations (Muller et al., 2005). Therefore, high Cu might be a general condition in this fjord due to the rainy nature
of the geographical location, and indigenous plankton might have developed physiological mechanisms to deal with
high Cu, such as the production of organic ligands to prevent uptake (Vraspir and Butler, 2009), or of heavy-metal-
binding peptides (phytochelatins) to lower Cu toxicity inside the cell (Ahner and Morel, 1995; Ahner et al., 1995;
Knauer et al., 1998). Since we measured high particulate Cu, and Cu:P in our experiment, *E. huxleyi* might have been
relying mainly on phytochelatins to buffer high intracellular Cu (Ahner et al., 2002).

The Cd:P ratios (average 0.024 ± 0.01 mmol Cd:mol P, Figure 2) were significantly lower than those in phytoplankton
and *E. huxleyi* (0.36 mmol Cd:mol P, Figure 2). This was surprising, because Cd quotas are normally higher in
coccolithophores than in diatoms and chlorophytes (Sunda and Huntsman, 2000; Ho et al., 2003). High Cd quotas in
coccolithophores have been suggested to result from accidental uptake through Ca transporters and channels (Ho et al.,
2009). The low Cd quotas here may be explained by the antagonistic interaction between Mn and Cd or Zn and Cd
under high Mn and Zn, respectively (Sunda and Huntsman, 1998, 2000; Cullen and Sherrell, 2005).
Since high Zn:P ratios were common in this study (34.02 ± 18.05 mmol Zn:mol P, Figure 2), we hypothesize that high
Zn levels antagonistically interacted with Cd, resulting in low Cd:P ratios in the particles.

**4.4 The effects of increased $CO_2$ and the DFB addition on particulate metal concentrations and P-normalized**
**ratios**
Fe enrichment is common in coastal waters, due to sediment resuspension, rivers input, aeolian deposition and mixing
or upwelling of deep water. Indeed, Fe was the essential metal with the highest particulate concentrations in our study
(Figure 1, Table 3). Furthermore, in this study particulate Fe was characterized by a strong lithogenic component, and
was not correlated with phytoplankton biomass. Fe was also unique, in that it was the only trace element whose
particulate concentration was significantly and     exclusively affected by $CO_2$     (no interaction between $CO_2$ and
DFB), regardless of the presence or absence of DFB (Table 1)     . Furthermore, particulate Fe concentrations (nM)
decreased steadily between days 12 and 21, with the exception of the control treatment (LC-DFB; Figure 1, Table 2S).
This suggests that the increase in $CO_2$ and/or the DFB addition reduce the concentration of pFe, despite the
phytoplankton bloom. Such a decrease in pFe (range 2.3-fold in LC-DFB, vs. 13.7-fold in HC+DFB; Table S2) might
be mediated by the dissolution of particulate Fe by low pH or by the presence of strong organic chelators as observed in
this very experiment (Segovia et al. 2017 and references therein) where dFe notably increased in treatments with high
$CO_2$ and/or the addition of DFB (Figure S3). Furthermore, the dissolution of particulate Fe in the treatments with high
$CO_2$ and/or the addition of DFB was evident in the Fe partitioning coefficients—the molar ratio between particulate and
dissolved concentrations (Figure 4). On day 21, the Fe partitioning coefficients varied by 22-fold between the highest
for the control (LC-DFB: 1.039) and lowest for the HC+DFB treatments (HC+DFB: 0.047; Figure 4). Thus, either the
DFB addition or high $CO_2$ promoted the dissolution of pFe. However, at the end of the experiment, high dFe
concentrations were only observed in the treatments with the DFB additions, suggesting that the presence of strong
organic Fe chelators, such as DFB, mediated the maintenance of high dissolved Fe concentrations, as previously
observed (Segovia et al. 2017). Thus, in our future oceans, high $CO_2$ (low pH) will increase dissolved Fe concentrations
in regions rich in particulate Fe, and in strong organic Fe chelators. The deleterious effects of OA on the development
of ecologically important species sensitive to increased $CO_2$ such as *E. Huxleyi*, will be more relevant in high-Fe
environments than in Fe-limited ones.
In contrast to the findings for Fe, particulate Cu concentrations a) were not affected by either high $CO_2$ or the DFB
addition; b) were dominated by a biogenic component and c) were significantly correlated with phytoplankton biomass
(Table 4). Furthermore, unique to Cu was a significant increase in Cu:P ratios by day 21 in the high $CO_2$ treatments,
especially when no DFB was added. Since the Cu partitioning coefficients only varied by 3.25 fold among treatments
on day 21 (LC-DFB: 0.065 vs. HC+DFB: 0.047; data not shown), we hypothesize that high $CO_2$ did not affect the
partitioning between particulate and dissolved Cu, but instead, it affected the speciation of dissolved Cu, increasing free
Cu ($Cu^{2+}$) and thus, its bioavailability. This resulted in the highest Cu:P ratios in the high $CO_2$ treatments, despite the
low phytoplankton biomass. This increase in bioavailability under lower pH is typical of metals that form strong
inorganic complexes with carbonates, such as $Cu^{2+}$ (Millero et al., 2009). Thus in our future oceans, high $CO_2$ (low pH)
will shift the speciation of dissolved Cu towards higher abundance of free ionic species, increasing its bioavailability
and likely its toxicity.
Similarly to Cu, particulate Co, Zn and Mn were correlated with biomass and were dominated by the biogenic
component. But in contrast to Cu, these metals particulate concentrations were affect by increased $CO_2$ and/or the DFB
addition. However, the effects of high $CO_2$ and/or DFB were very complex because significant interactions between
these 2 factors were observed (Table 1); and further studies are required before we are able to discern and conclude a
significant trend. Yet, the P-normalized ratios of Co, Zn and Mn were significantly affected by $CO_2$ (Table 3),
exhibiting moderately lower ratios under high $CO_2$, when phytoplankton biomass was lowest. These results imply that
the bioavailability of these metals was not enhanced under acidic conditions. This suggests that under high $CO_2$ (low
pH) the free ionic species of these metals will not significantly increase in the future, as shown for metals that occur
predominantly as free ionic species in seawater (Millero et al., 2009).

**5. Concluding remarks**
The results presented here show that in the fjord where we carried out the present experiment, particulate Fe was
dominated by lithogenic material, and was significantly decreased in the treatments with high $CO_2$ concentrations
and/or the DFB addition. Indeed, high $CO_2$ and/or DFB promoted the dissolution of particulate Fe, and the presence
of this strong organic complex helped maintaining high dissolved Fe. This shift between particulate and dissolved
Fe, in the presence of DFB, promoted a massive bloom of *E. huxleyi* in the treatments with ambient $CO_2$, due to
increased dissolved Fe. During the bloom of *E. huxleyi*, the concentrations of particulate metals (mol $L^{-1}$) with
a strong biogenic component (Cu, Co, Zn, Cd, Mn, and Mo) were a) highly dynamic, b) positively correlated with
plankton biomass, and c) influenced by growth requirements. Furthermore, high $CO_2$ decreased the Me:P ratios of Co,
Zn and Mn, while increased the Cu:P ratios. In contrast DFB had no effects on these ratios. According to our results,
high $CO_2$ may decrease particulate Fe and increase dissolved Fe, but high concentrations of dissolved Fe will only be
maintained by the presence of strong organic ligands. Furthermore, ocean acidification will decrease *E. huxleyi*
abundance, and as a result, the sinking of particulate metals enriched in *E. huxleyi*. Moreover, the Me:P ratios of metals
that are predominately present in an ionic free form in seawater (e.g. Co, Zn and Mn) will likely decrease or stay
constant. In contrast, high $CO_2$ is predicated to shift the speciation of dissolved metals associated with carbonates, such
as Cu, increasing their bioavailability, and resulting in higher Me:P ratios. We suggest that high Cu might be a
common condition in this fjord, and autochthonous plankton might be able to cope with high Cu levels by developing
specific physiological mechanisms. Future high $CO_2$ levels are expected to change the relative concentrations of
particulate and dissolved metals, due to the differential effects of high $CO_2$ on trace metal solubility, speciation,
adsorption and toxicity, as well as on the growth of different phytoplankton taxa, and their elemental trace metal
composition.

**Acknowledgments**
This work was funded by CTM/MAR 2010-17216 (PHYTOSTRESS) research grant from the Spanish Ministry for
Science and Innovation (Spain) to MS, and by NSERC grants (Canada) to MTM and JTC. MRL was funded by a FPU
grant from the Ministry for Education (Spain) and by fellowships associated to the mentioned above research grants to
carry out a short-stay at MTM and JTC laboratories to analyze dissolved and particulate metals. We thank all the

participants of the PHYTOSTRESS experiment for their collaboration, and the MBS (Espegrend, Norway) staff for logistic support during the experiment. We thank the anonymous reviewers for insightful comments and constructive criticisms.

**Conflict of interest**

Authors declare no conflict of interest

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

**Table 1.** Statistical analyses (Split-plot ANOVA) of the effects of high $CO_2$, the addition of DFB, and their interaction, as well as the effect of time, on the concentrations of particulate metals (mmol $L^{-1}$, data in Table S2, and Figure 3) in particles collected from the different mesocosms treatments. We used all the days for the analyses because the Split-Plot ANOVA integrates fixed factors (Co2 and Fe) and a repeated measures factor (time) by using the post-hoc Bonferroni, thus, time was fully considered during the whole experimental period.

| Factor | Al | Ti | P | Fe | Cu | Co | Zn | Cd | Mn | Mo | Pb |
|---|---|---|---|---|---|---|---|---|---|---|---|
| $CO_2$ | ns | ns | ** | * | ns | ** | *** | *** | ** | *** | ns |
| DFB | ns | ns | * | ns | ns | * | ** | ns | * | * | ns |
| $CO_2$ x DFB | ns | * | ** | ns | ns | * | ** | * | ** | ** | ns |
| Time | ns | ns | ns | *** | * | *** | *** | *** | *** | *** | ** |

*ns: not significant; * p <0.05; ** p <0.01; *** p<0*

**Table 2.** The average metal ratios in the particles collected in this study (without oxalate wash) using the data reported in Table S2. The P-normalized ratios (mmol : mol P, Figure 4) are compared to previous estimates in marine plankton samples and phytoplankton cultures (A). The Al-normalized ratios (mmol : mol Al) are compared to crustal ratios (B).

A)

| (mmol : mol P) | Mn:P | Fe:P | Co:P | Cu:P | Zn:P | Cd:P | Mo:P | Al:P | Reference |
|---|---|---|---|---|---|---|---|---|---|
| Phytoplankton Lab | 3.8 | 7.5 | 0.19 | 0.38 | 0.8 | 0.21 | 0.03 | | Ho et al. 2003 |
| Marine Plankton Field | 0.68±0.54 | 5.1±1.6 | 0.15±0.06 | 0.41±0.16 | 2.1±0.88 | | | | Ho 2006 |
| *E. huxleyi* Lab | 7.1±0.36 | 3.5±0.07 | 0.29±0.02 | 0.07±0.013 | 0.38±0.002 | 0.36±0.01 | 0.022±0.0003 | | Ho et al. 2003 |
| **This study** | 1.65±0.41 | 39.2±34.3 | 0.07±0.02 | 1.41±0.55 | 34.02±18.05 | 0.02±0.01 | 0.42±0.12 | 70±38 | |
| Crust ratio | 510 | 29,738 | 13 | 25 | 32 | 0.05 | 0.46 | 89,972 | Taylor 1964 |

B)

| (mmol : mol Al) | Mn:Al | Fe:Al | Co:Al | Cu:Al | Zn:Al | Cd:Al | Mo:Al | Pb:Al | Ti:Al |
|---|---|---|---|---|---|---|---|---|---|
| Crustal ratio | 5.7 | 331 | 0.14 | 0.27 | 0.35 | 0.001 | 0.005 | 0.02 | 39 |
| **This study** | 35±28 | 506±342 | 1.5±1.2 | 26.5±15 | 795±865 | 0.5±0.4 | 8.6±6.5 | 4.9±3.9 | 119±47.6 |

**Table 3.** Statistical analyses (Split-plot ANOVA) of the effects of $CO_2$, DFB, and their interaction, as well as the effect of time, on the P-normalized metal quotas (mmol: mol P, data in Figure 4, and Table S2) in particles collected from the different mesocosm treatments.

| Factor | Fe:P | Cu:P | Co:P | Zn:P | Cd:P | Mn:P | Mo:P | Pb:P | Ti:P |
|---|---|---|---|---|---|---|---|---|---|
| $CO_2$ | ns | * | *** | ** | ns | * | ns | ns | ns |
| DFB | ns | ns | ns | ns | ns | ns | ns | ns | ns |
| $CO_2$ x DFB | ns | ns | ns | ns | ns | ns | ns | ns | ns |
| Time | *** | *** | *** | *** | ns | ns | ns | ns | *** |

*ns: not significant; * p <0.05; ** p <0.01; *** p<0.001*

**Table 4.** The relationship (Pearson correlations, $p < 0.05$) between particulate metals concentrations (nmol L$^{-1}$, no oxalate wash, reported in Table S2) and the biomass ($\mu$gC L$^{-1}$) of *Emiliania huxleyi* and total cells (phytoplankton and microzooplankton) collected from the different mesocosm treatments.

| | | **P** | **Fe** | **Cu** | **Co** | **Zn** | **Cd** | **Mn** | **Mo** | **Pb** | **Ti** |
|---|---|---|---|---|---|---|---|---|---|---|---|
| *E. huxleyi* | Correlation coefficient | 0.622 | ns | 0.614 | 0.756 | 0.747 | 0.818 | 0.686 | 0.825 | ns | ns |
| | P-value | 0.003 | | 0.003 | $7.35 \cdot 10^{-5}$ | $1.01 \cdot 10^{-4}$ | $6.02 \cdot 10^{-6}$ | $5.93 \cdot 10^{-4}$ | $4.20 \cdot 10^{-6}$ | | |
| Total cells | Correlation coefficient | 0.641 | ns | 0.51 | 0.644 | 0.889 | 0.802 | 0.598 | 0.53 | ns | ns |
| | P-value | 0.002 | | 0.02 | $1.62 \cdot 10^{-3}$ | $7.03 \cdot 10^{-8}$ | $1.23 \cdot 10^{-5}$ | $4.18 \cdot 10^{-3}$ | $1.35 \cdot 10^{-2}$ | | |

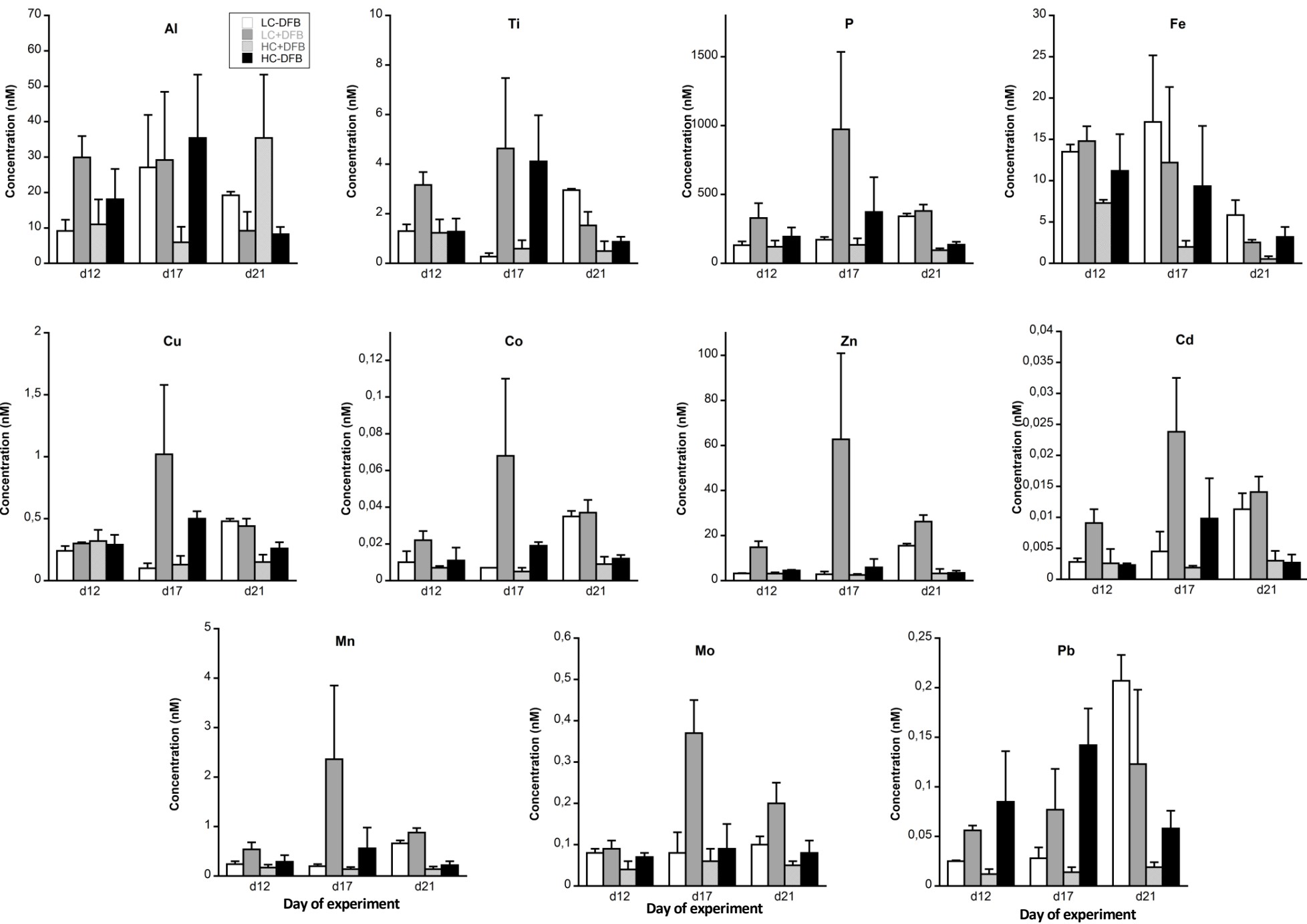

**Fig. 1**. The concentration of particulate metals in seawater (nM) in the different treatments; LC: ambient $CO_2$ (390 µatm); HC: increased $CO_2$ (900 µatm); -DFB (ambient dFe); +DFB (increased dFe) during the development of a bloom of Emiliania huxleyi. Bars are means of measurements in 3 independent mesocosms (n = 3) except for LC−DFB where n = 2. Error bars indicate SD.

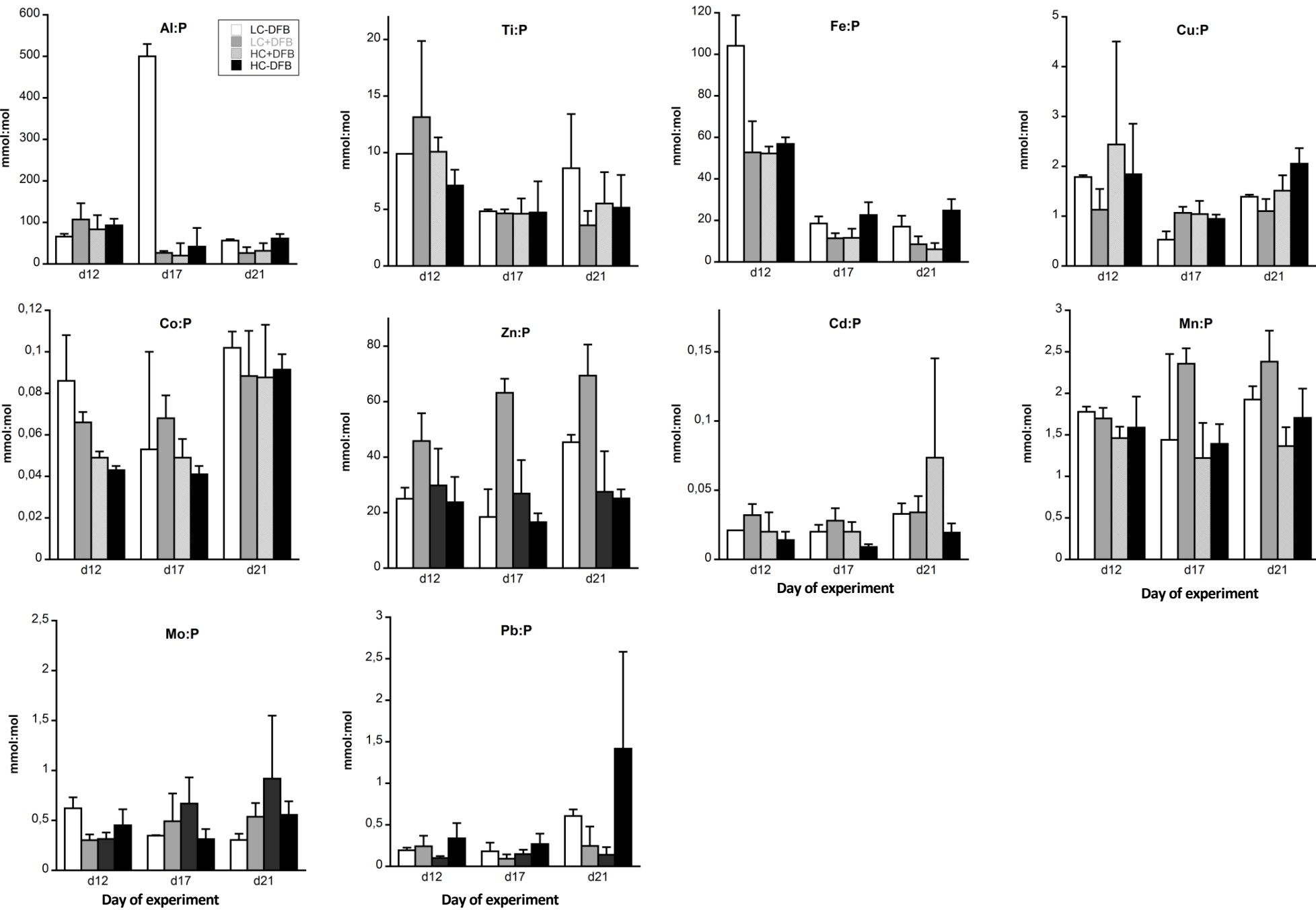

**Fig. 2**. P-normalized metal quotas (mmol:mol P) of particles from different treatments; LC: ambient $CO_2$ (390 µatm); HC: increased $CO_2$ (900 µatm); -DFB (ambient dFe); +DFB (increased dFe) during the development of a bloom of *Emiliania huxleyi*. Bars are means of measurements in 3 independent mesocosms (n = 3) except for LC−DFB where n = 2. Error bars indicate SD.

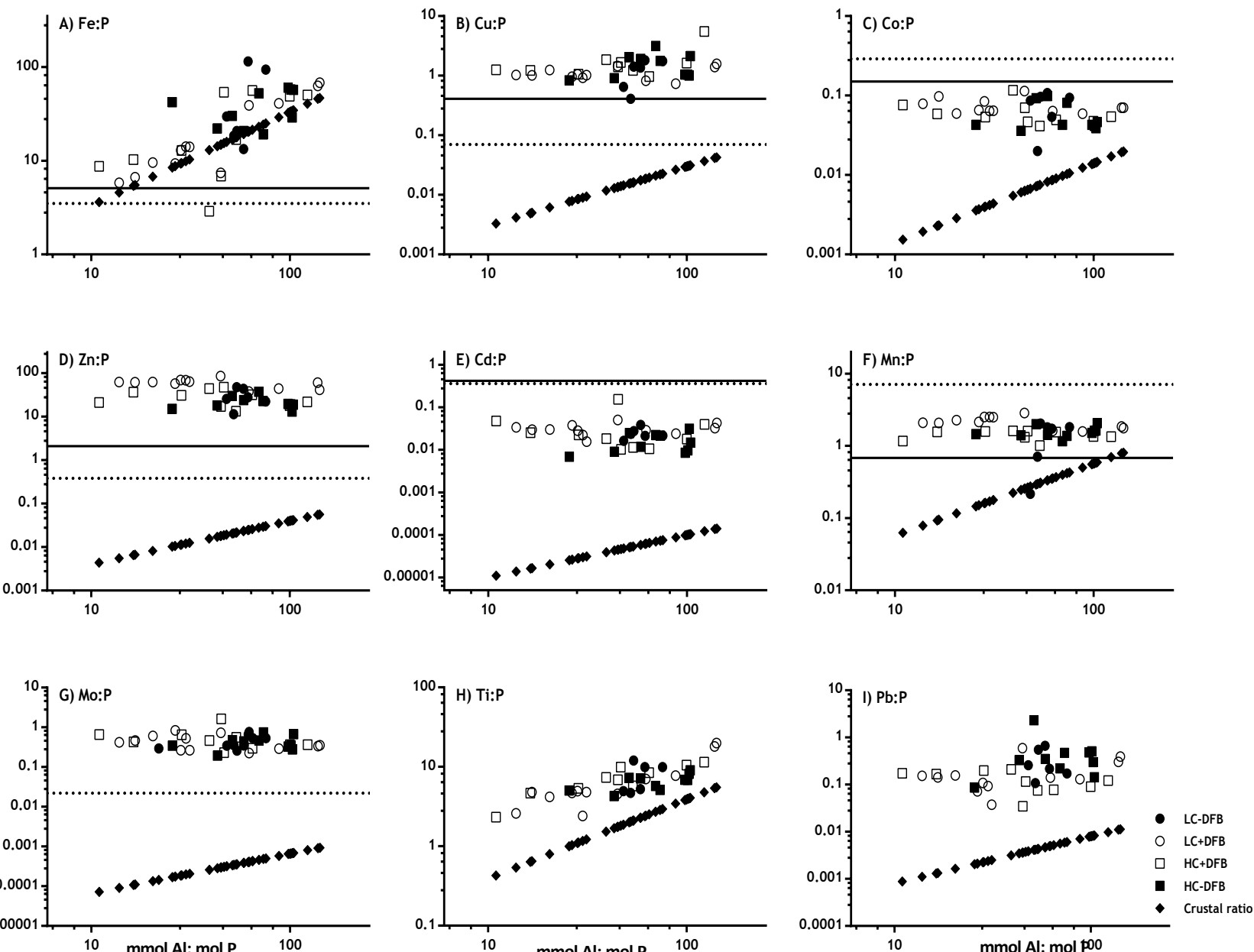

**Fig. 3**. Comparison of P-normalized metal ratios in particles (mmol:mol P) against mmol Al:mol P ratios in the same particles (without oxalate wash) collected from the different mesocosm treatments (LC: ambient $CO_2$; HC: increased $CO_2$ (900 µatm); -DFB: no DFB addition; +DFB: with a 70 nM DFB addition) during the development of a bloom on day 12, 17 and 21 (original data reported in Table S2). The x-axis parallel solid and dotted lines represent the average metal quotas obtained from marine plankton assemblages (Ho 2006) and from cultures of *Emiliania huxleyi* (Ho et al. 2003). The slope of the line with the ♦ symbols indicates the average metal : Al (mol:mol) in crustal material (Taylor, 1964). (A) Fe:P, (B) Cu:P, (C) Co:P, (D) Zn:P, (E) Cd:P, (F) Mn:P, (G) Mo:P, (H) Ti:P, (I) Pb:P.

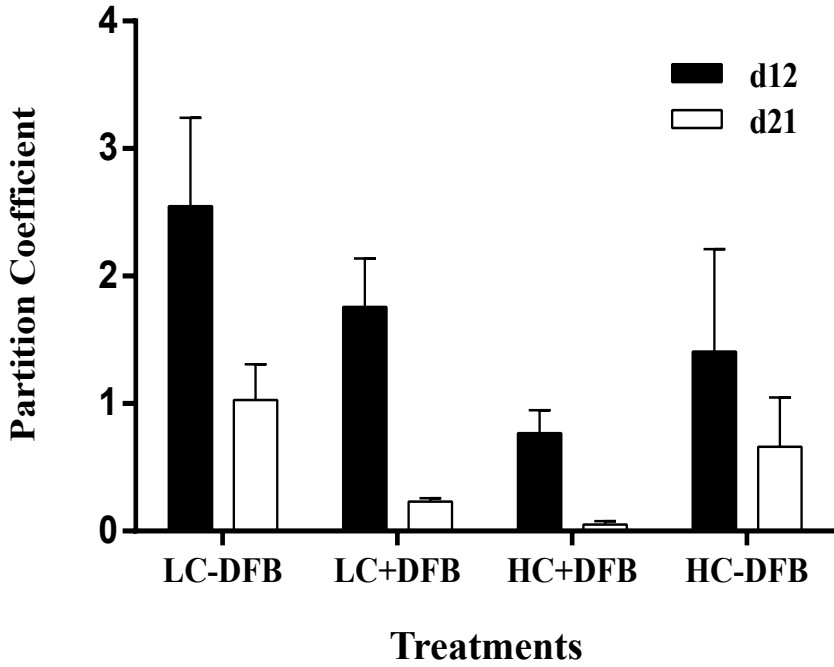

**Fig. 4**. The Fe partition coefficients (the molar ratio between particulate and dissolved concentrations) in the different mesocosm treatments; LC: ambient $CO_2$ (390 μatm); HC: increased $CO_2$ (900 μatm); -DFB: no DFB addition; +DFB: with a 70 nM DFB addition; on day 12 and day 21. . Bars are means of measurements in 3 independent mesocosms (n = 3) except for LC−DFB where n = 2. Error bars indicate SD.