# Peer review of "Particulate trace metal dynamics in response to increased CO2"

_Biogeosciences, 2018_

## Referee Comment (RC1) · Anonymous Referee #1 · 15 Dec 2018

Review comments for Biogeochemical Discuss Particulate trace metal dynamics in response to increased CO2 and iron availability in a coastal mesocosm experiment Authors: M. R. Lorenzo, M. Segovia, J. T. Cullen, and M. T. Maldonado

This manuscript reports a particulate trace metal data set in controlled mesocosm experiments which conducted in the Raunefiord. Four conditions were prepared for the mesocosms and particulate trace metal concentrations were measured during Emiliania huxleyi blooming which induced by artificially. The authors claimed that they found the results that marine particle trace metal is highly dynamic, positively correlated with phytoplankton biomass, influenced by growth requirement, and strongly affected by

changing CO2 level and Fe availability.

Generally, the topic of the change of particle trace metal during the marine environment changing such as Ocean acidification (OA) and different Fe availability is very interesting for ocean biogeochemsits. I believe the data set is valuable in this field. I feel that, however, authors need to regard more about how they can present their data set to induce conclusion above, which they claimed in conclusion section in this paper. The present contents of this manuscript are not well organized for presenting their data set to conclude the claimed conclusions.

General comments In construction of this manuscript, "results" section is not constructed only by result, and "discussion" section is not well explained by this study's result (data) ("Discussions" are only like a review of previous knowledge). I recommend that authors should re-construct and re-organized whole part of the manuscript. "Results" section should be used some "Figures" for presenting their data. It makes more easily to understand for readers. "Discussion" section should be related more to data from this study, including which data induce which conclusion more clearly.

The effect of CO2 did not follow a clear trend in this study, as authors mentioned in the text. The effect of controlled Fe availability by DFB addition/non-addition to phytoplankton bloom is also not clear. How authors induced these their claimed results is not clearly understandable for readers. For discussion OA influence, I think authors should focus on to show "How particle trace metal concentrations and its ratio changed by CO2 concentrations" by more well presented their own data set. For Fe availability, they need to discuss that "did DFB addition influence positive/negative to Fe availability?". It depends on natural dissolved Fe concentrations. Additionally, authors should show more clearly about relationship between Fe availability and E. huxleyi bloom response, with figure etc. It is very difficult for readers to understand the relation only from the "Tables" number. DFB addition inducing more dissolved fraction of TM is artificial response. This is different story from Fe bio-availability. Important for bio-availability is how much free Fe exist under each condition. As one of author well know that DFB-Fe
uptake by phytoplankton need very complexed mechanisms. Authors should discuss more detail about this part.

Specific comments Line 86ïïjŽAuthors describe "a bloom of the coccolithophorid Emiliania huxleyi was induced in a mesocosm experiment". This manuscript described time variation of particle trace metal parameter on Day 12 (d12), d17, d21. It is better to show the bloom development and termination in each mesocosm, with particle trace metal concentrations and other data, by Figures. This is more helpful to readers for understanding what was occurring in the each mesocosm. Authors can show such as cell number, chlorophyll a, POC, nutrients conc. for basic information for back grand of this study. If it is already reported, part of this can be cited from published work.

Line 116-117:, 171 Table 2 etc: Authors described that "The biological and chemical variables analyzed were phytoplankton abundance and species composition, dissolved Fe and Cu, nutrient concentration, and particle trace metals concentration". They only show these data in Tables. Figures which present time variation are easier for readers to understand the data variation during the experiments. Please prepare Figures. I can imagine the particle trace metal data was only collected on d12, d17, d21. But for grasp biological response and chemical environment change, sampling should be done more frequently. If authors have more frequent data for nutrient, cell number for E. huxleyi, etc, it should be plotted to the Figures.

3. Results Line 170- , 3.1 Biological and chemical characteristics during bloom., 178-181: Authors described time changing of "diatom" with nutrient concentrations. Authors should make a plot of "day since day0" vs "diatom cell number", vs "pigment", vs "E. huxlei" and vs "nutrient concentrations" in each mesocosm. It is helpful for reader.

Line 188-189, 3.2: Authors described "The only metal that showed a time-dependent decrease in its particulate concentrations was Fe. . . . . .. . . .. . .. . . . ."dMetals", "pMetals". These also should be appeared by figures.

176: Authors described "An increase in dFe was observed in all treatments between

day 7 and 17.". In Table 2, there are no data from d7, it should be appeared in the Table 2. And d17 LC-DFB data is decrease. So "all treatments" in this sentence is not correct.

Line 195-196: Authors indicate that "high CO2 had negative impact on particulate Fe" and "Cd concentrations were also inversely affected by CO2". These parts are difficult to understand which data indicate this fact.

Line 197-198: Authors described "All other elements (P, Co. . . . . . . . . .. . ...these two factors was clear (Table 5)". Please explain how clear as like Cd description in previous sentence. Only showing Table 5 is not enough explanation to reader.

For "3. Results" section, all subtitle is not well organized. Some contents can be compiled to one (For example, 3.1 and 3.2 can merge for "biological chemical response in mesocosms". And particle trace meatal variation in different treatment in 3.2, 3.3, 3.4 can merge to one section.

Title and contents of subsection in 3.6 and 3.7 are part of "discussion".

4.Discussion Line 247-248: "Our results demonstrate that in the studied fjord, particulate Ti and Fe concentrations were dominated by lithogenic material.". Authors need explanation how they judged this. The explanation is appeared in section 3.6 result (actually this is discussion). Please indicate clearly "this data is shown in Figure 1".

Line248-250: "In contrast, particulate Cu, Co, Mn, Zn, Mo and Cd concentrations were correlated with P concentrations, as well as phytoplankton biomass, suggesting strong biogenic influence on their distribution (Table 6)". Authors need explanation how they judged this. The explanation is appeared in section 3.7 result (actually this is discussion). Please indicate clearly "this data is shown in Figure 1". Only showing Table 6 is not kind for reader. This part is overlapped to 4.2 section. It should be in to 4.2 section with detailed explanations with Tables and figures.

Line 251-252: "Changes in CO2 and/or Fe levels affected total particulate and biogenic

metal concentrations for some metals.". This part of results is not well presented in manuscript overall. Authors should regard to present some figures which can compare particle and biogenic metals concentrations among each treatment.

Line 255-263, 4.1 Efficacy of the oxalate-EDTA wash removing lithogenic trace metals from particles: First half part of this section is should be move to "results". Especially from line 260-263, "In general, the concentrations of Fe and Co in the particles were decreased the least by the oxalate wash (by $\sim$ 25%), while Mo and Pb concentrations were decreased the most (by $\sim$70%). The concentrations of particulate Cu, Zn, Cd and Mn were reduced by 50% by the oxalate wash. As shown previously (Sanudo-Wilhelmy et al. 2004), the oxalate reagent also removed extracellular P (by $\sim$20%).".

Line 282:" Me:P ratios we measured in the particles are similar to those of natural phytoplankton assemblages (Ho, 2006) and of Emiliania huxleyi cultures (Ho et al., 2003).". If authors want to compare their filed data to previous reported data by Ho, 2006, and Ho et al., 2003, authors should show the previous study's number with their data on to Tables or Figures with citation. Otherwise, authors just state "similar" to natural plankton but did not show any evidence.

Line 311-312, 319-320: "Interestingly, we also found a putative ZIP-transporter gene. ZIP-transporters are. . . . . . . . . . ., such as tRNA synthetase, reverse transcriptase, metallo-carboxypeptidase, ABC-Zn-transporter and CDF-Zn-transporter..". If authors want to say "we found", they should show their data and discuss with using their data. If this "gene part" is part of other study, they should cite the other study appropriately. This discussion section is very strange for this aspect. It is written like author's original data for this study.

Line 329- : Discussion on Cu:P should construct by using their data, what their data's characteristics, what their data indicate, what is authors claim from the data, which previous knowledge supports their claims. This section 4.2 is like just a review of other papers.

Line 344: "The Cd:P were significantly lower than those found in phytoplankton and E. huxleyi.". Reader can not understand clearly which data they compared. Is this sentence mean that "The Cd:P were significantly lower than those found in individual phytoplankton and E. huxleyi which was reported by previous studies (Ho, 2006, Ho et al., 2003)"?. If so, they should show the comparable data from previous study.

Line 377: "The decrease in particulate Fe might have been due to enhanced solubility of Fe- oxides at low pH.". The author should show scientific basis. They have to show relation between pFe and PH in each treatment.

Line 378: "the concentration of the elements P, Co, Zn, Mn and Mo were influenced by CO2 and Fe levels". Which data indicate those results? Authors should present with their dataset.

Line 380-381: "where the addition of DFB resulted in higher dissolved Fe, and optimal pH enhanced E. huxleyi growth.". Authors should present this relation, between dissolved Fe, pH and E. huxleyi growth, with figures which are constructed by their dataset.

5. conclusion remarks Please consider for my "general comment". It is necessary to describe more specifically what was understood in each argument (claim) a)-d).

Authors should present what are difference/similarity of their data among four mesocosms treatment more clearly, and what they can find from the difference/similarity? How they induced the conclusion of this study form the difference/similarity? This aspect is not clear overall in this manuscript.

Others Authors used "pFe", "particulate Fe", "particulate iron", "dFe", and "dissolved Fe" in the text. They should use same words through the manuscript.

End of review.

---

## Referee Comment (RC2) · Anonymous Referee #2 · 24 Jan 2019

Review for

Particulate trace metal dynamics in response to increased CO2 and iron availability in a coastal mesocosm experiment

by Lorenzo and colleagues

General comments. This study examines the changes in particulate trace metal concentrations in response to CO2 and Fe manipulations in a coastal mesocosm experiment. While the data of the study are interesting and research of this kind is very important and should be encouraged, the quality of the manuscript needs to be improved considerably before it is suitable for publication. I found it very hard to follow the

description of the data in the Results, as most of them have been presented in tables, which is especially not good for presenting the time dependent changes in for example trace metal concentrations. I also found that the Discussion for the most part was on trace metal chemistry and physiology in general, but not specifically relative to the key objective of the study, i.e., the effects of CO2 and Fe availability on particulate trace metal dynamics.

Specific comments. Line 91. "(Hutchings, 2011)", which was not included in the References.

Line 108. 10 uM nitrate: 0.3 uM PO4 = 33:1 - was there a particular reason to use such a P limited nutrient condition? How may this affect the observed particulate trace metal concentration?

Line 109. what is the ratio of Fe to DFB? 1:1?

Line 137. Oxalate-EDTA wash can remove not only extracellular Fe, but also other metals. Change "...extracellular Fe" to "...extracellular metals".

Results: I would strongly suggest that the data should be presented as figures instead of tables. In addition, albeit statistical analyses were conducted and presented together in Table 5, I would suggest they should also be presented in each individual table (or figure, if the authors decide to follow my suggestion above in revising the manuscript).

Lines 170-181. "days 1-10, phase I", "day 7" and "After day 10" were mentioned when describing the data, but none of them can be found in Table 2.

Line 221. "(Figure 5)" should be Table 5.

Discussion: Again the Discussion mostly did not center around the influence of acidification and/or Fe availability on trace metal dynamics, except for the last, very short section 4.4. I thus encourage the authors to considerably revise the Discussion, focusing on how the chemistry and utilization of trace metals were affected by changes in CO2/pH and Fe levels and how these may be related to the proliferation of Ehux in the

mesocosm.

Lines 366-367. Should be Table 2, not Table 1.

---

## Author Comment (AC1) · 16 May 2019

RESPONSE TO REVIEWERS Anonymous Referee #1 1)... Generally, the topic of the change of particle trace metal during the marine environment changing such as Ocean acidification (OA) and different Fe availability is very interesting for ocean bio-geochemists. I believe the data set is valuable in this field. I feel that, however, authors need to regard more about how they can present their data set to induce conclusion above, which they claimed in conclusion section in this paper. The present contents of this manuscript are not well organized for presenting their data set to conclude the claimed conclusions. General comments: In construction of this manuscript, "results"

section is not constructed only by result, and "discussion" section is not well explained by this study's result (data) ("Discussions" are only like a review of previous knowledge). I recommend that authors should re-construct and re-organized whole part of the manuscript. "Results" section should be used some "Figures" for presenting their data. It makes more easily to understand for readers. "Discussion" section should be related more to data from this study, including which data induce which conclusion more clearly. In view of both reviewers comments we have substantially changed the Ms attending to their requierements. The abstract has been modified, as well as the results and discussion sections. We would like to remark that specifically the Discussion section has been fully re-structured and re-discussed. We believe it is more focused now in order to get good conclusions. Specific answers to the comments raised follow below. 2)The effect of CO2 did not follow a clear trend in this study, as authors mentioned in the text. The effect of controlled Fe availability by DFB addition/non-addition to phytoplankton bloom is also not clear. How authors induced these their claimed results is not clearly understandable for readers. This has been modified accordingly in 2.1 Experimental set-up section, Lns 119-124: "To induce changes in Fe availability, and analyse its effects on the plankton community, 70 nM (final concentration) of the siderophore desferrioxamine B (DFB) (+DFB and -DFB treatments) (Figure 1b) was added to half of the mesocosms on Day 7, when the community was already acclimated to high CO2. The initial dFe concentration before DFB addition was about 4.5 nM. Even though DFB is a strong Fe-binding organic ligand often used to induce iron limitation in phytoplankton (Wells 1999), DFB additions may also increase the dissolved Fe pool in environments with high concentrations of colloidal and/or particulate Fe, such as fjords (Kuma et al. 1996, Öztürk et al. 2002). By day 17, dissolved iron concentrations were significantly higher (by ~3-fold) in the high CO2 and DFB treatments than in the control (Segovia et al. 2017). These results support an increase in the solubility of Fe in seawater by either lowering its pH (Millero 1998; Millero et al. 2009) and/or the addition of DFB (Chen et al. 2004)." We have added Figure 1, comprising 3 panels referring to CO2 concentration, dFe and dCu concentrations to better

understand the chemical scenario that lead the experiment to the obtained results. 3) For discussion OA influence, I think authors should focus on to show "How particle trace metal concentrations and its ratio changed by CO2 concentrations" by more well presented their own data set. We have changed the tables by bar plots (now figures 3 and 4) according to reviewer's suggestions and we think it is much clearer now. We have re-written section 3.3-The effects of increased CO2 and the DFB addition on particulate metal concentrations. Lns 233-242. "Increased CO2 and the DFB addition did not significantly affect the concentrations of particulate AI, Ti, Cu, and Pb (Tables 1 and S2). Similarly, the addition of DFB did not directly influence particulate concentrations of Fe, but high CO2 had a significant negative impact on particulate Fe (Tables 1 and S2, Figure 3). Particulate Cd concentrations were also inversely affected by CO2, but only in the presence of DFB (CO2; and CO2 x DFB effect, Tables 1 and S2, Figure 3). All other elements (P, Co, Zn, Mn and Mo) exhibited significant effects by CO2 and by DFB, but there was also a significant interaction between these two factors (Table 1, S2). This indicates that, for example, particulate Mn, Zn, Mo, Co, and P concentrations were significantly decreased by high CO2, but only in the +DFB treatments (Figure 3, Table 1, S2,). Similarly, the addition of DFB significantly increased pZn and pMn, but only at ambient CO2 levels (Figure 3, Table 1, S2)" Data are now supported by new Fig 3, Table 1 and Suppl.Table S2. 4) For Fe availability, they need to discuss that "did DFB addition influence positive/negative to Fe availability?". It depends on natural dissolved Fe concentrations. See answer to point 2 above, and point 5 below. 5) Additionally, authors should show more clearly about relationship between Fe availability and E. huxleyi bloom response, with figure etc. It is very difficult for readers to understand the relation only from the "Tables" number. We have substituted the Table by a new figure 2 with permission from Segovia et al. Mar. Ecol. Prog. Ser. 2017 for a better understanding. This figure shows the temporal development of chlorophyll a ( $\mu$ g L-1) and phytoplankton biomass ( $\mu q C L - 1$ ) in the mesocosms exposed to different CO2 and dissolved iron (dFe) treatments. (a) Chlorophyll a, (b) Emiliania huxleyi (5–10  $\mu$ m), (c) Synechococcus  $(0.6-2 \mu m)$ , (d) picoeukaryotes  $(0.1-2 \mu m)$ , (e) small nanoeukary-

СЗ

otesl(prasinophytes, small haptophytes, 2-7  $\mu$ m), (f) large nanoeukaryotes (small single-celled diatoms and flagellated forms, 6-20 µm),ĺ(g) diatoms (chain-forming Skeletonema sp. 20-> 500  $\mu$ m), (h) dinoflagellates (20-200  $\mu$ m). This is also discussed in a new 4.1 Section: "The effects of CO2 and dFe in the plankton community In this experiment we investigated changes in particulate trace metal concentrations, in response to increased CO2 and/or an addition of the siderophore DFB in a coastal mesocosm experiment. For a better understanding of the processes affecting these stressors, we briefly summarise the mesocosm experiment results originating from Segovia et al. (2017). High CO2, as well as the DFB addition increased dFe concentration. The higher dFe concentrations were sustained in the DFB treatments. A bloom of the coccolithophore Emiliania huxleyi was observed in the ambient CO2 treatments, and was especially massive in the one with the addition of DFB (LC+DFB). On the contrary, the biomass of E. huxleyi was negatively affected by increased CO2. However, increased dFe partially mitigated the negative effect of elevated CO2, indicating that the coccolithophore was able to acclimate better to ocean acidification when Fe availability was high. High dFe also had a positive effect on the cyanobacterium Synechococcus sp, while the rest of the plankton food web did not response to the treatments (Segovia et al. 2017)" 6) DFB addition inducing more dissolved fraction of TM is artificial response. This is different story from Fe bio-availability. Important for bio-availability is how much free Fe exist under each condition. As one of author well know that DFB-Fe uptake by phytoplankton need very complex mechanisms. Authors should discuss more detail about this part. This was fully discussed in Segovia et al. 2017 and it is not the aim of this paper which is focused in pMe. However, we suggest to consult Segovia et .2017 for further details. See Lns 423-427. "Under control conditions at present CO2 concentration (LC, 380 µatm) and no DFB amendment, the globally important coccolithophore Emiliania huxleyi was experiencing Fe limitation (Segovia et al. 2017). The shift between particulate and dissolved Fe promoted a massive bloom of E. huxleyi in the treatments with ambient CO2 due to increased Fe bioavailability (for further details on Fe-bioavailability in E. huxleyi please see Segovia et a. 2017)".

7)Line 116-117:, 171 Table 2 etc.: Authors described that "The biological and chemical variables analysed were phytoplankton abundance and species composition, dissolved Fe and Cu, nutrient concentration, and particle trace metals concentration". They only show these data in Tables. Figures which present time variation are easier for readers to understand the data variation during the experiments. Please prepare Figures. I can imagine the particle trace metal data was only collected on d12, d17, and d21. But for grasp biological response and chemical environment change, sampling should be done more frequently. If authors have more frequent data for nutrient, cell number for E. huxleyi, etc., it should be plotted to the Figures. Authors described time changing of "diatom" with nutrient concentrations. Authors should make a plot of "day since day0" vs. "diatom cell number", vs. "pigment", vs. "E. huxlei" and vs. "nutrient concentrations" in each mesocosm. It is helpful for reader.

It is indeed done. We have included Chla and phytoplankton biomass in the new Fig.2. Dissolved Fe and Cu in Fig. 1, and Nitrate, Ammonium, Silicate and Phosphate in new Supplemental Figure S1. Additionally, readers are encouraged to consult Segovia et al. 2017. There is no point of re-publishing already published results. Only those that are really essential (our opinion). 8)Line 188-189, 3.2: Authors described "The only metal that showed a time-dependent decrease in its particulate concentrations appeared by figures. 176: Authors described "An increase in dFe was observed in all treatments between day 7 and 17.". In Table 2, there are no data from d7, it should be appeared in the Table 2. And d17 LC-DFB data is decrease. So "all treatments" in this sentence is not correct. This is already solved by the new figures and the explanation in the results section. In addition, Supplemental Figure S2 shows the Fe partition coefficients (the molar ratio between particulate and dissolved concentrations) in the different mesocosm treatments; LC: ambient CO2 (390 µatm); HC: increased CO2 (900  $\mu$ atm); -DFB: no DFB addition; +DFB: with a 70 nM DFB addition; on day 12 and day. 9)Line 195-196: Authors indicate that "high CO2 had negative impact on particulate Fe" and "Cd concentrations were also inversely affected by CO2". These parts are

difficult to understand which data indicate this fact. Line 197-198: Authors described "All other elements (P, Co. . . . . . . . . . . . . . . . . . these two factors was clear (Table 5)". Please explain how clear as like Cd description in previous sentence. Only showing Table 5 is not enough explanation to reader. We have rewritten 3.3-The effects of increased CO2 and the DFB addition on particulate metal concentrations, as follows: "Increased CO2 and the DFB addition did not significantly affect the concentrations of particulate AI, Ti, Cu, and Pb (Tables 1 and S2). Similarly, the addition of DFB did not directly influence particulate concentrations of Fe, but high CO2 had a significant negative impact on particulate Fe (Tables 1 and S2, Figure 3). Particulate Cd concentrations were also inversely affected by CO2, but only in the presence of DFB (CO2; and CO2 x DFB effect, Tables 1 and S2, Figure 3). All other elements (P, Co, Zn, Mn and Mo) exhibited significant effects by CO2 and by DFB, but there was also a significant interaction between these two factors (Table 1, S2). This indicates that, for example, particulate Mn, Zn, Mo, Co, and P concentrations were significantly decreased by high CO2, but only in the +DFB treatments (Figure 3, Table 1, S2,). Similarly, the addition of DFB significantly increased pZn and pMn, but only at ambient CO2 levels (Figure 3, Table 1, S2)" We believe it is much clearer now. Note that Table 5 has been substituted by figure 3 to make it easier, and that Table 1 and S2 also support these data. 10)For "3. Results" section, all subtitle is not well organized. Some contents can be compiled to one (For example, 3.1 and 3.2 can merge for "biological chemical response in mesocosms". And particle trace metal variation in different treatment in 3.2, 3.3, and 3.4 can merge to one section. Some of the sub-headings have been reorganised. We have accepted the suggestion of "3.1 biological chemical response in mesocosms". However, we have not merged the following sections as we think it will be rather confusing to the reader. 11)Title and contents of subsection in 3.6 and 3.7 are part of "discussion". These results parts have been re-structured and part of the text has been moved to the Discussion section. 12) Discussion Line 247-248: "Our results demonstrate that in the studied fjord, particulate Ti and Fe concentrations were dominated by lithogenic material.". Authors need explanation how they judged this. The explanation is appeared in section 3.6 result (actually this is discussion). Please indicate clearly "this data is shown in Figure 1". Line248-250: "In contrast, particulate Cu, Co, Mn, Zn, Mo and Cd concentrations were correlated with P concentrations, as well as phytoplankton biomass, suggesting strong biogenic influence on their distribution (Table 6)". Authors need explanation how they judged this. The explanation is appeared in section 3.7 result (actually this is discussion). Please indicate clearly "this data is shown in Figure 1". Only showing Table 6 is not kind for reader. This part is overlapped to 4.2 section. It should be in to 4.2 section with detailed explanations with Tables and figures. This is one of the parts that has changed most. This section is supported by Fig 5 and Table 4. Thus, we have fully restructured and re-discussed 4.2 Particulate Fe and Ti are associated with lithogenic sources, while particulate Co, Cu, Zn, Cd, Mo and Mn are associated with biogenic sources, as follows, Lns 265-326: "The particulate trace metal concentrations (nM, mean of all treatments and dates) during the experiment were highest for AI, Fe and Zn, and lowest for Co and Cd, following this trend: Al  $\approx$  Fe  $\approx$  Zn > Ti > Cu  $\approx$  Mn > Mo  $\approx$  Pb > Co > Cd. Lithogenic particles are enriched in AI and low in P (average crustal AI and P content is 2.9 mmol AI and 0.034 mmol P g-1 dry weight, Taylor 1964), while biogenic particles are enriched in P and low in AI (average plankton AI and P content is 0.001 mmol AI and 0.26 mmol P g-1 dry weight, Bruland et al. 1991). Therefore, the distinct high abundance of Al and P in lithogenic and biogenic particles, respectively, can be used to evaluate the relative contribution of lithogenic and biogenic material in our particulate samples. In order to do this, first, it is important to establish that the vast majority of the measured particulate P is associated the biogenic fraction. In this study, the abiotic P was estimated using particulate AI concentrations (nM) and the P:AI ratio in crustal material, and was calculated to be negligible (

Time (days of experiment) Fig. 1. Temporal development of (a) CO2 partial pressure (pCO2) and (b) pH within the mesocosms. Ambient pCO2 and anicreased dFe (HC-0FB, red ), and (anicreased dFe (LC-0FB, red filled circle); increased pCO2 and increased dFe (HC-0FB, red open circle), increased pCO2 and ambient dFe (HC-0FB, lack open circle). Symbols indicate means of measurements in 3 independent mesocosms (n = 3) except for LC-0FB where n = 2. Error bars indicate SD. Figure reproduced with permission from Segovia et al. Mar. Ecol. Prog. Ser. 2017

Fig. 1. Figures

527Table 1. Statistical analyses (Split-plot ANOVA) of the effects of high CO2, the addition of DFB, and their interaction, as well as the effect of528time, on the concentrations of particulate metals (mmol  $L^4$ , data in Table S2, and Figure 3) in particles collected from the different mesocosms529treatments.

|     | Factor                                                    | Al | Ti | Р  | Fe  | Cu | Co  | Zn  | Cd  | Mn  | Mo  | Pb |
|-----|-----------------------------------------------------------|----|----|----|-----|----|-----|-----|-----|-----|-----|----|
|     | CO 2                                           | ns | ns | ** | *   | ns | **  | *** | *** | **  | *** | ns |
|     | DFB                                                       | ns | ns | *  | ns  | ns | *   | **  | ns  | *   | *   | ns |
|     | CO2 x DFB                                                 | ns | *  | ** | ns  | ns | *   | **  | *   | **  | **  | ns |
|     | Time                                                      | ns | ns | ns | *** | *  | *** | *** | *** | *** | *** | ** |
| 530 | 0 ns: not significant; * p <0.05; ** p <0.01; *** p<0.001 |    |    |    |     |    |     |     |     |     |     |    |

Fig. 2. Tables

---

## Author Comment (AC2) · 16 May 2019

Referee #2

I found it very hard to follow the description of the data in the Results, as most of them have been presented in tables, which is especially not good for presenting the time dependent changes in for example trace metal concentrations. I also found that the Discussion for the most part was on trace metal chemistry and physiology in general, but not specifically relative to the key objective of the study, i.e., the effects of CO2 and Fe availability on particulate trace metal dynamics. See point 1, 21 and others to Referee #1 Specific comments. Line 91. "(Hutchings, 2011)", which was not

included in the References. Now included Line 108. 10 uM nitrate: 0.3 uM PO4 = 33:1 - was there a particular reason to use such a P limited nutrient condition? How may this affect the observed particulate trace metal concentration? Yes, we used this specific ratio because we aimed at an Emiliania huxleyi bloom. This has been clarified as follows, Lns 117-119: At the beginning of the experiment, nitrate (10 $\mu$M final concentration) and phosphate (0.3 $\mu$M final concentration) were added to induce a bloom of the coccolithophore Emiliania huxleyi, as recommended by Egge & Heimdal (1994). Results: I would strongly suggest that the data should be presented as figures instead of tables. In addition, albeit statistical analyses were conducted and presented together in Table 5, I would suggest they should also be presented in each individual table (or figure, if the authors decide to follow my suggestion above in revising the manuscript). Done Lines 170-181. "days 1-10, phase I", "day 7" and "After day 10" were mentioned when describing the data, but none of them can be found in Table 2. Line 221. "(Figure 5)" should be Table 5. This has now changed with the new Ms organisation. Discussion: Again the Discussion mostly did not center around the influence of acidification and/or Fe availability on trace metal dynamics, except for the last, very short section 4.4. I thus encourage the authors to considerably revise the Discussion, focusing on how the chemistry and utilization of trace metals were affected by changes in CO2/pH and Fe levels and how these may be related to the proliferation of Ehux in the mesocosm. This has now changed with the new Ms organisation. See comment to Referee 1. We thank the reviewers for their comments and their time, and hope that our responses are satisfactory Yours sincerely, Maria Segovia & Maite Maldonado

Please also note the supplement to this comment:
https://www.biogeosciences-discuss.net/bg-2018-448/bg-2018-448-AC2-supplement.pdf
* * *
[Figure]

[Figure]

[Figure]

**Fig. 1.** Temporal development of (a) CO2 partial pressure (pCO2) and (b) pH within the mesocosms. Ambient pCO2 and ambient dFe (LC−DFB, grey); ambient pCO2 and increased dFe (LC+DFB, red filled circle); increased pCO2 and increased dFe (HC+DFB, red open circle), increased pCO2 and ambient dFe (HC−DFB, black open circle). Symbols indicate means of measurements in 3 independent mesocosms (n = 3) except for LC−DFB where n = 2. Error bars indicate SD. Figure reproduced with permission from Segovia et al. Mar. Ecol. Prog. Ser. 2017

**Fig. 1.** Figures

527 **Table 1.** Statistical analyses (Split-plot ANOVA) of the effects of high $CO_2$, the addition of DFB, and their interaction, as well as the effect of
528 time, on the concentrations of particulate metals (mmol $L^{-1}$, data in Table S2, and Figure 3) in particles collected from the different mesocosms
529 treatments.

| Factor | Al | Ti | P | Fe | Cu | Co | Zn | Cd | Mn | Mo | Pb |
|---|---|---|---|---|---|---|---|---|---|---|---|
| $CO_2$ | ns | ns | ** | * | ns | ** | *** | *** | ** | *** | ns |
| DFB | ns | ns | * | ns | ns | * | ** | ns | * | * | ns |
| $CO_2$ x DFB | ns | * | ** | ns | ns | * | ** | * | ** | ** | ns |
| Time | ns | ns | ns | *** | * | *** | *** | *** | *** | *** | ** |

530 *ns: not significant; * p <0.05; ** p <0.01; *** p<0.001*

**Fig. 2.** Tables

---

## Author Response (AR1)

RESPONSE TO REVIEWERS on "Particulate trace metal dynamics in response to increased CO2 and iron availability in a coastal mesocosm experiment" *by* M. Rosario Lorenzo et al.

**Anonymous Referee #1**

1)... Generally, the topic of the change of particle trace metal during the marine environment changing such as Ocean acidification (OA) and different Fe availability is very interesting for ocean biogeochemists. I believe the data set is valuable in this field. I feel that, however, authors need to regard more about how they can present their data set to induce conclusion above, which they claimed in conclusion section in this paper. The present contents of this manuscript are not well organized for presenting their data set to conclude the claimed conclusions.

General comments: In construction of this manuscript, "results" section is not constructed only by result, and "discussion" section is not well explained by this study's result (data) ("Discussions" are only like a review of previous knowledge). I recommend that authors should re-construct and re-organized whole part of the manuscript. "Results" section should be used some "Figures" for presenting their data. It makes more easily to understand for readers. "Discussion" section should be related more to data from this study, including which data induce which conclusion more clearly.

In view of both referee comments we have substantially change the Ms attending to their comments. Abstract has been modified, as well as the results and discussion sections. We would like to pinpoint that specifically the Discussion section has been fully re-structured and re-discussed. We believe it is more focused now in order to get a good conclusion. Specific answers to the comments raised follow below.

**2)The effect of CO2 did not follow a clear trend in this study, as authors mentioned in the text. The effect of controlled Fe availability by DFB addition/non-addition to phytoplankton bloom is also not clear. How authors induced these their claimed results is not clearly understandable for readers.**

This has been modified accordingly in 2.1 Experimental set-up section, Lns 119-124: "To induce changes in Fe availability, and analyse its effects on the plankton community, 70 nM (final concentration) of the siderophore desferrioxamine B (DFB) (+DFB and -DFB treatments) (Figure 1b) was added to half of the mesocosms on Day 7, when the community was already acclimated to high CO2. The initial dFe concentration before DFB addition was about 4.5 nM. Even though DFB is a strong Fe-binding organic ligand often used to induce iron limitation in phytoplankton (Wells 1999), DFB additions may also increase the dissolved Fe pool in environments with high concentrations of colloidal and/or particulate Fe, such as fjords (Kuma et al. 1996, Öztürk et al. 2002). By day 17, dissolved iron concentrations were significantly higher (by  $\sim$ 3-fold) in the high CO2 and DFB treatments than in the control (Segovia et al. 2017). These results support an increase in the solubility of Fe in seawater by either lowering its pH (Millero 1998; Millero et al. 2009) and/or the addition of DFB (Chen et al. 2004)."

We have added Figure 1, comprising 3 panels referring to  $CO_2$  concentration, dFe and dCu concentrations to better understand the chemical scenario that lead the experiment to the obtained results.

**3) For discussion OA influence, I think authors should focus on to show "How particle trace metal concentrations and its ratio changed by CO2 concentrations" by more well presented their own data set.**

We have changed the tables by bar plots (now figures 3 and 4) according to reviewer's suggestions and we think it is much clearer now. We have re-written section 3.3-The effects of increased CO2 and the DFB addition on particulate metal concentrations. Lns 233-242. "Increased CO2 and the DFB addition did not significantly affect the concentrations of particulate Al, Ti, Cu, and Pb (Tables 1 and S2). Similarly, the addition of DFB did not directly influence particulate concentrations of Fe, but high CO2 had a significant negative impact on particulate Fe (Tables 1 and S2, Figure 3). Particulate Cd concentrations were also inversely affected by CO2, but only in the presence of DFB (CO2; and CO2 x DFB effect, Tables 1 and S2, Figure 3). All other elements (P, Co, Zn, Mn and Mo) exhibited significant effects by CO2 and by DFB, but there was also a significant interaction between these two factors (Table 1, S2). This indicates that, for example, particulate Mn, Zn, Mo, Co, and P concentrations were significantly decreased by high CO2, but only in the +DFB treatments (Figure 3, Table 1, S2,). Similarly, the addition of DFB significantly increased pZn and pMn, but only at ambient CO2 levels (Figure 3, Table 1, S2)."

Data are now supported by new Fig 3, Table 1 and Suppl.Table S2.

**4) For Fe availability, they need to discuss that "did DFB addition influence positive/negative to Fe availability?". It depends on natural dissolved Fe concentrations.**

See answer to point 2 above, and point 5 below.

**5) Additionally, authors should show more clearly about relationship between Fe availability and E. huxleyi bloom response, with figure etc. It is very difficult for readers to understand the relation only from the "Tables" number.**

We have substituted the Table by a new figure 2 with permission from Segovia et al. Mar. Ecol. Prog. Ser. 2017 for better understanding. This figure shows the temporal development of chlorophyll a ( $\mu$ g L-1) and phytoplankton biomass ( $\mu$ g C L-1) in the mesocosms exposed to different CO2 and dissolved iron (dFe) treatments. (a) Chlorophyll a, (b) Emiliania huxleyi (5–10  $\mu$ m), (c) Synechococcus (0.6–2  $\mu$ m), (d) picoeukaryotes (0.1–2  $\mu$ m), (e) small nanoeukaryotes (prasinophytes, small haptophytes, 2–7  $\mu$ m), (f) large nanoeukaryotes (small single-celled diatoms and flagellated forms, 6–20  $\mu$ m), (g) diatoms (chain-forming Skeletonema sp. 20–> 500  $\mu$ m), (h) dinoflagellates (20–200  $\mu$ m).

This is also discussed in a new 4.1 Section: "The effects of  $CO_2$  and dFe in the plankton community

In this experiment we investigated changes in particulate trace metal concentrations, in response to increased  $CO_2$  and/or an addition of the siderophore DFB in a coastal mesocosm experiment. For a better understanding of the processes affecting these stressors, we briefly summarise the mesocosm experiment results originating from Segovia et al. (2017). High  $CO_2$ , as well as the DFB addition increased dFe concentration. The higher dFe concentrations were sustained in the DFB treatments. A bloom of the coccolithophore Emiliania huxleyi was observed in the ambient  $CO_2$  treatments, and was especially massive in the one with the addition of DFB (LC+DFB). On the contrary, the biomass of E. huxleyi was negatively affected by increased  $CO_2$ . However, increased dFe partially mitigated the negative effect of elevated  $CO_2$ , indicating that the coccolithophore was able to acclimate better to ocean acidification when Fe availability was high. High dFe also had a positive effect on the cyanobacterium Synechococcus sp, while the rest of the plankton food web did not response to the treatments (Segovia et al. 2017).

6) DFB addition inducing more dissolved fraction of TM is artificial response. This is different story from Fe bio-availability. Important for bio-availability is how much free Fe exist under each condition. As one of author well know that DFB-Fe uptake by phytoplankton need very complex mechanisms. Authors should discuss more detail about this part.

This was fully discussed in Segovia et al. 2017 and it is not the aim of this paper which is focused in pMe. However, we suggest to consult Segovia et .2017 for further details. See Lns 423-427. "Under control conditions at present  $CO_2$  concentration (LC, 380 µatm) and no DFB amendment, the globally important coccolithophore Emiliania huxleyi was experiencing Fe limitation (Segovia et al. 2017). The shift between particulate and dissolved Fe promoted a massive bloom of E. huxleyi in the treatments with ambient  $CO_2$  due to increased Fe bioavailability (for further details on Fe-bioavailability in E. huxleyi please see Segovia et al. 2017)".

7)Line 116-117:, 171 Table 2 etc.: Authors described that "The biological and chemical variables analysed were phytoplankton abundance and species composition, dissolved Fe and Cu, nutrient concentration, and particle trace metals concentration". They only show these data in Tables. Figures which present time variation are easier for readers to understand the data variation during the experiments. Please prepare Figures. I can imagine the particle trace metal data was only collected on d12, d17, and d21. But for grasp biological response and chemical environment change, sampling should be done more frequently. If authors have more frequent data for nutrient, cell number for E. huxleyi, etc., it should be plotted to the Figures. Authors described time changing of "diatom" with nutrient concentrations. Authors should make a plot of "day since day0" vs. "diatom cell number", vs. "pigment", vs. "E. huxlei" and vs. "nutrient concentrations" in each mesocosm. It is helpful for reader.

It is indeed done. We have included Chla and phytoplankton biomass in the new Fig.2. Dissolved Fe and Cu in Fig. 1, and Nitrate, Ammonium, Silicate and Phosphate in new Supplemental Figure S1. Additionally reader are encouraged to consult Segovia et al. 2017.

There is no point of re-publishing already published results. Only those that are really essential

This is already solved by the new figures and the explanation in results section. In addition we have included Supplemental Figure S2 showing the Fe partition coefficients (the molar ratio between particulate and dissolved concentrations) in the different mesocosm treatments; LC: ambient CO2 (390 µatm); HC: increased CO2 (900 µatm); -DFB: no DFB addition; +DFB: with a 70 nM DFB addition; on day 12 and day.

We have rewritten 3.3-The effects of increased  $CO_2$  and the DFB addition on particulate metal concentrations as follows: "Increased  $CO_2$  and the DFB addition did not significantly affect the concentrations of particulate Al, Ti, Cu, and Pb (Tables 1 and S2). Similarly, the addition of DFB did not directly influence particulate concentrations of Fe, but high  $CO_2$  had a significant negative impact on particulate Fe (Tables 1 and S2, Figure 3). Particulate Cd concentrations were also inversely affected by  $CO_2$ , but only in the presence of DFB ( $CO_2$ ; and  $CO_2 \times DFB$  effect, Tables 1 and S2, Figure 3). All other elements (P, Co, Zn, Mn and Mo) exhibited significant effects by  $CO_2$  and by DFB, but there was also a significant interaction between these two factors (Table 1, S2). This indicates that, for example, particulate Mn, Zn, Mo, Co, and P concentrations were significantly decreased by high  $CO_2$ , but only in the +DFB treatments (Figure 3, Table 1, S2,). Similarly, the addition of DFB significantly increased pZn and pMn, but only at ambient  $CO_2$  levels (Figure 3, Table 1, S2)."

We believe is much clearer now. Note that Table 5 has been substituted by figure 3 to make it easier and that Table 1 and S2 also support these data.

**10)For "3. Results" section, all subtitle is not well organized. Some contents can be compiled to one (For example, 3.1 and 3.2 can merge for "biological chemical response in mesocosms". And particle trace metal variation in different treatment in 3.2, 3.3, and 3.4 can merge to one section.**

Some of he sub-headings have reorganised. We have accepted the suggestion of "3.1 *biological chemical response in mesocosms*". However, we have not merged the following sections as we think it will be rather confusing to the reader.

**11)Title and contents of subsection in 3.6 and 3.7 are part of "discussion".**

These results parts have been re-structured and part of the text has been moved to Discussion section.

12) Discussion Line 247-248: "Our results demonstrate that in the studied fjord, particulate Ti and Fe concentrations were dominated by lithogenic material.". Authors need explanation how they judged this. The explanation is appeared in section 3.6 result (actually this is discussion). Please indicate clearly "this data is shown in Figure 1". Line248-250: "In contrast, particulate Cu, Co, Mn, Zn, Mo and Cd concentrations were correlated with P concentrations, as well as phytoplankton biomass, suggesting strong biogenic influence on their distribution (Table 6)". Authors need explanation how they judged this. The explanation is appeared in section 3.7 result (actually this is discussion). Please indicate clearly "this data is shown in Figure 1". Only showing Table 6 is not kind for reader. This part is overlapped to 4.2 section. It should be in to 4.2 section with detailed explanations with Tables and figures.

This one of the parts that have changed most. This section is supported by Fig 5 and Table 4. Thus, we have fully restructured and re-discussed 4.2 
[revised manuscript text omitted]

**13)Line 251-252: "Changes in CO2 and/or Fe levels affected total particulate and biogenic metal concentrations for some metals.". This part of results is not well presented in manuscript overall. Authors should regard to present some figures which can compare particle and biogenic metals concentrations among each treatment.**

We have included Figure 3 and 4, and eliminated the corresponding tables that were unclear.

14)Line 255-263, 4.1 Efficacy of the oxalate-EDTA wash removing lithogenic trace metals from particles: First half part of this section is should be move to "results". Especially from line 260-263, "In general, the concentrations of Fe and Co in the particles were decreased the least by the oxalate wash (by ~ 25%), while Mo and Pb concentrations were decreased the most (by ~70%). The concentrations of particulate Cu, Zn, Cd and Mn were reduced by 50% by the oxalate wash. As shown previously (Sanudo-Wilhelmy et al. 2004), the oxalate reagent also removed extracellular P (by ~20%).".

This has been moved to material and methods and now reads as follows. Lns 176-195:

**"2.3.3 The effect of oxalate-EDTA wash on particulate trace metal concentrations**

To better estimate the biogenic fraction of the particulate metals, the filters were washed with an oxalate-EDTA solution, which removes extracellular metals and oxyhydroxides (Tovar-Sanchez et al., 2003; Tang and Morel, 2006). In our study, the oxalate wash significantly decreased the concentration of all particulate metals, with the exception of Al and Ti (Tables S2 and S3), as observed by Rauschenberg and Twining (2015). The quantity of metal remaining after the oxalate wash (i.e. biogenic fraction) varied among elements (Tables S2 and S3). In general, the concentrations of Fe and Co in the particles were decreased the least by the oxalate wash by ~ 25%, while Mo and Pb concentrations were decreased the most by ~70%. The concentrations of particulate Cu, Zn, Cd and Mn were reduced by 50% by the oxalate wash. As shown previously (Sanudo-Wilhelmy et al. 2004), the oxalate reagent also removed extracellular P (by ~20%, Table S2 & S3). Compared to Rauschenberg and Twining (2015), the estimates of the biogenic fraction, after the oxalate wash, were in agreement for Co, Cu and P, and lower for Fe, Mn, Zn and Cd concentrations.

However, the efficacy of the oxalate wash to dissolve Fe, and other metals, from lithogenic particles is not well constrained (Frew et al. 2006, Rauschenberg and Twining, 2015, King et al., 2012). Therefore, the results obtained after the oxalate-EDTA wash should be interpreted with caution because we do not know whether the removed metal fraction is a) only lithogenic; b) mainly lithogenic but some biogenic fraction is also removed, or c)

whether metals absorbed onto particles are equally labile to the wash on biogenic and lithogenic particles. Given that many of the trends we observed were identical for the oxalate-EDTA washed and non-washed particles [i.e. higher Me concentrations in the LC+DFB treatments (Table S2 & S3) and positive correlations between phytoplankton biomass and Me concentrations (Lorenzo-Garrido 2016)], below we present and discuss only the non-oxalate wash results"

So that, oxalate-wash is not further discussed. Yet, it is important to maintain this paragraph in the Ms.

15) Line 282:" Me:P ratios we measured in the particles are similar to those of natural phytoplankton assemblages (Ho, 2006) and of Emiliania huxleyi cultures (Ho et al., 2003).". If authors want to compare their filed data to previous reported data by Ho, 2006, and Ho et al., 2003, authors should show the previous study's number with their data on to Tables or Figures with citation. Otherwise, authors just state "similar" to natural plankton but did not show any evidence.

This has all been re-organised and re-discussed as required by both reviewers. Please see point 12 above.

16) Line 311-312, 319-320: "Interestingly, we also found a putative ZIP-transporter gene. ZIP-transporters are. . . . . . . . . . , such as tRNA synthetase, reverse transcriptase, metallo-carboxypeptidase, ABC-Zn-transporter and CDF-Zn-transporter..". If authors want to say "we found", they should show their data and discuss with using their data. If this "gene part" is part of other study, they should cite the other study appropriately. This discussion section is very strange for this aspect. It is written like author's original data for this study.

We have removed the paragraph relating to referee comments. We fully agree with the comment.

17) Line 329- : Discussion on Cu:P should construct by using their data, what their data's characteristics, what their data indicate, what is authors claim from the data, which previous knowledge supports their claims. This section 4.2 is like just a review of other papers.

We have re-written this part and better discussed our own data as follows, Lns 352-362:

Similarly, the Cu:P ratios in the collected particles were relatively elevated  $(1.4 \pm 0.8 \text{ mmol } \text{Cu: mol } P)$  compared to those of other phytoplankton, including E. huxleyi (Table 2). The dissolved  $(7.7\pm0.41 \text{ nM } \text{Cu}, \text{Figure } 1)$  and particulate Cu concentrations  $(0.35\pm0.25 \text{ nM}, \text{Table } S2)$  in our experiment were high, and similar to those previously measured in this fjord (Muller et al., 2005). Rain events (or wet deposition of anthropogenic aerosols) in this fjord result in high dissolved Cu and the active production of strong organic ligands by Synechococcus—to lower the free Cu concentrations (Muller et al., 2005). Therefore, high Cu might be a general condition in this fjord, and indigenous plankton might have developed physiological mechanisms to deal with high Cu, such as the production of organic ligands to prevent uptake (Vraspir and Butler, 2009), or of heavy-metal-binding

peptides (phytochelatins) to lower Cu toxicity inside the cell (Ahner and Morel, 1995; Ahner et al., 1995; Knauer et al., 1998). Since we measured high particulate Cu, and Cu:P in our experiment, E. huxleyi might have been relying mainly on phytochelatins to buffer high intracellular Cu (Ahner et al., 2002).

18) Line 344: "The Cd:P were significantly lower than those found in phytoplankton and E. huxleyi.". Reader can not understand clearly which data they compared. Is this sentence mean that "The Cd:P were significantly lower than those found in individual phytoplankton and E. huxleyi which was reported by previous studies (Ho, 2006, Ho et al., 2003)"?. If so, they should show the comparable data from previous study.

We have clarified this in section 4.3 Particulate metals with a strong biogenic component: their P-normalized ratios, Lns 364-372 :" *The Cd:P ratios (average 0.024 \pm 0.01 mmol Cd:mol P, Figure 4 and 6) were significantly lower than those in phytoplankton and E. huxleyi (0.36 mmol Cd:mol P, Figure 4 and 6). This was surprising, because Cd quotas are normally higher in coccolithophores than in diatoms and chlorophytes (Sunda and Huntsman, 2000; Ho et al., 2003). High Cd quotas in coccolithophores have been suggested to result from accidental uptake through Ca transporters and channels (Ho et al., 2009). The low Cd quotas here may be explained by the antagonistic interaction between Mn and Cd or Zn and Cd under high Mn and Zn, respectively (Sunda and Huntsman, 1998, 2000; Cullen and Sherrell, 2005). Since high Zn:P ratios were common in this study (34.02 ± 18.05 mmol Zn:mol P, Figure 4 and 6), we hypothesize that high Zn levels antagonistically interacted with Cd, resulting in low Cd:P ratios in the particles"*

**19)Line 377: "The decrease in particulate Fe might have been due to enhanced solubility of Fe- oxides at low pH.". The author should show scientific basis. They have to show relation between pFe and PH in each treatment.**

It is now shown in Figure 1 and Figure S2 and discussed in 4.4 The effects of increased  $CO_2$  and the DFB addition on particulate metal concentrations and P-normalized ratios, Lns 373-395:

"Iron enrichment is common in coastal waters, due to sediment resuspension, rivers input, aeolian deposition and mixing or upwelling of deep water. Indeed, Fe was the essential metal with the highest particulate concentrations in our study (Figure 3, Table 3). Furthermore, in this study particulate Fe was characterized by a strong lithogenic component, and was not correlated with phytoplankton biomass. Iron was also unique, in that it was the only trace element whose particulate concentration was significantly and uniquely affected by CO2, regardless of the presence or absence of DFB (no interaction between CO2 and DFB, Table 1). Furthermore, particulate Fe concentrations (nM) decreased steadily between days 12 and 21, with the exception of the control treatment (LC-DFB; Figure 3, Table 2S). This suggests that the increase in  $CO_2$  and/or the DFB addition reduce the concentration of pFe, despite the phytoplankton bloom. Such a decrease in pFe (range 2.3-fold in LC-DFB, vs. 13.7-fold in HC+DFB; Table S2) might be mediated by the dissolution of particulate Fe by low pH or by the presence of strong organic chelators as observed in this very experiment (Segovia et al. 2017 and references therein). where dFe notably increased in treatments with high CO2 and/or the addition of DFB (Figure 1). Furthermore, the dissolution of particulate Fe in the treatments with high

 $CO_2$  and/or the addition of DFB was evident in the Fe partitioning coefficients—the molar ratio between particulate and dissolved concentrations. On day 21, the Fe partitioning coefficients varied by 22-fold between the highest for the control (LC-DFB: 1.039) and lowest for the HC+DFB treatments (HC+DFB: 0.047; Figure S2). Thus, either the DFB addition or high  $CO_2$  promoted the dissolution of pFe. However, at the end of the experiment, high dFe concentrations were only observed in the treatments with the DFB additions, suggesting that the presence of strong organic Fe chelators, such as DFB, mediated the maintenance of high dissolved Fe concentrations, as previously observed (Segovia et al. 2017). Thus, in our future oceans, high  $CO_2$  (low pH) will increase dissolved Fe concentrations in regions rich in particulate Fe, and in strong organic Fe chelators."

20)Line 378: "the concentration of the elements P, Co, Zn, Mn and Mo were influenced by CO2 and Fe levels". Which data indicate those results? Authors should present with their dataset. Line 380-381: "where the addition of DFB resulted in higher dissolved Fe, and optimal pH enhanced E. huxleyi growth.". Authors should present this relation, between dissolved Fe, pH and E. huxleyi growth, with figures which are constructed by their dataset.

This is now solved by inclusion of Fig 1, Fig 2, by changing Tables to Figs 3 and 4 and answered in points 3,6,9 and 19.

21) conclusion remarks Please consider for my "general comment". It is necessary to describe more specifically what was understood in each argument (claim) a)-d). Authors should present what are difference/similarity of their data among four mesocosms treatment more clearly, and what they can find from the difference/similarity? How they induced the conclusion of this study form the difference/similarity? This aspect is not clear overall in this manuscript.

We believe that this requirement is now met attending to the deep-structural changes we have done in the ms.

22) Others Authors used "pFe", "particulate Fe", "particulate iron", "dFe", and "dissolved Fe" in the text. They should use same words through the manuscript.

Changed accordingly

**Anonymous Referee #2**

I found it very hard to follow the description of the data in the Results, as most of them have been presented in tables, which is especially not good for presenting the time dependent changes in for example trace metal concentrations. I also found that the Discussion for the most part was on trace metal chemistry and physiology in general, but not specifically relative to the key objective of the study, i.e., the effects of CO2 and Fe availability on particulate trace metal dynamics.

See point 1, 21 and others to Referee #1

Specific comments. Line 91. "(Hutchings, 2011)", which was not included in the References.

Now included

**Line 108. 10 uM nitrate: 0.3 uM PO4 = 33:1 - was there a particular reason to use such a P limited nutrient condition? How may this affect the observed particulate trace metal concentration?**

Yes, we used this specific ratio because we aimed at a Emiliania huxleyi bloom. This has been clarified as follows, Lns 117-119:

At the beginning of the experiment, nitrate (10  $\mu$ M final concentration) and phosphate (0.3  $\mu$ M final concentration) were added to induce a bloom of the coccolithophore Emiliania huxleyi, as recommended by Egge & Heimdal (1994).

Results: I would strongly suggest that the data should be presented as figures instead of tables. In addition, albeit statistical analyses were conducted and presented together in Table 5, I would suggest they should also be presented in each individual table (or figure, if the authors decide to follow my suggestion above in revising the manuscript).

Done

Lines 170-181. "days 1-10, phase I", "day 7" and "After day 10" were mentioned when describing the data, but none of them can be found in Table 2. Line 221. "(Figure 5)" should be Table 5.

This has now changed with the new Ms organisation.

Discussion: Again the Discussion mostly did not center around the influence of acidification and/or Fe availability on trace metal dynamics, except for the last, very short section 4.4. I thus encourage the authors to considerably revise the Discussion, focusing on how the chemistry and utilization of trace metals were affected by changes in CO2/pH and Fe levels and how these may be related to the proliferation of Ehux in the mesocosm.

This has now changed with the new Ms organisation. See comment to Referee 1.

We thank the reviewers for their comments and their time, and hope that our responses are satisfactory

Yours sincerely,

Maria Segovia & Maite Maldonado

| Particulate trace metal dynamics in response to increased CO 2                                                                                 |
|-----------------------------------------------------------------------------------------------------------------------------------------------------------|
| and iron availability in a coastal mesocosm experiment                                                                                                    |
|                                                                                                                                                           |
| M. Rosario Lorenzo 1 , María Segovia 1 , Jav T. Cullen 2 , and María T. Maldonado 3                           |
|                                                                                                                                                           |
| 1 Department of Ecology, Faculty of Sciences, University of Málaga, Buleyar Louis Pasteur s/n, 29071-Málaga, Spain                             |
| 2 School of Earth and Ocean Sciences, University of Victoria, 3800 Finnerty Road, Bob Wright Centre, A405, Victoria                            |
| BC V8P 5C2                                                                                                                                                |
| Canada                                                                                                                                                    |
| 3 Department of Earth, Ocean and Atmospheric Sciences, University of British Columbia, 2207 Main Mall, Vancouver                               |
| BC V6T 1Z4, Canada                                                                                                                                        |
|                                                                                                                                                           |
| Correspondence to: María Segovia (segovia@uma.es) and María T. Maldonado (mmaldonado@coas.ubc.ca)                                                         |
|                                                                                                                                                           |
| Abstract. Rising concentrations of atmospheric carbon dioxide are causing ocean acidification and will influence                                          |
| marine processes and trace metal biogeochemistry. In June 2012, in Raunefjord (Bergen, Norway) we performed a                                             |
| mesocosm experiment, comprised of a fully factorial design of ambient and elevated pCO2 and/or an addition of the                                         |
| siderophore desferrioxamine B (DFB). In addition, the macronutrient concentrations were manipulated to enhance a                                          |
| bloom of the coccolithophore Emiliania huxleyi. We report here the changes in particulate trace metal (pMe)                                               |
| concentrations during this experiment. Our results show that particulate Ti and Fe were dominated by lithogenic                                           |
| material while particulate Cu, Co, Mn, Zn, Mo and Cd had a strong biogenic component. Furthermore, significant                                            |
| correlations were found between particulate concentrations (mol L-1) of Cu, Co, Zn, Cd, Mn, Mo, and P in seawater an                                      |
| phytoplankton biomass (µgC L-1), supporting a significant influence of the bloom in the distribution of these particulat                                  |
| elements. The concentrations of these biogenic metals (mol L -1 ) in the $E_{\mathbf{x}}$ huxleyi bloom were ranked as: $Zn > Cu \approx Mn$   |
| $\geq$ Mo > Co > Cd. Changes in CO 2 and/or DFB affected total particulate concentrations (mol L -1 ) and biogenic metal            |
| ratios (Me:P) for some metals. Variations in CO2 had the most clear, and significant effect on particulate Fe                                             |
| concentrations (mol L -1 ), decreasing its concentration under high CO 2 , Similarly, high CO 2 decreased the Co, Zn and |
| Mn: P ratios, while increased the Cu: P ratios. In contrast, the addition of DFB had no significant effect on any of the                                  |
| biogenic metal ratios, whilst high concentrations of dissolved Fe will only be maintained by the presence of strong                                       |
| organic ligands. Future predicted high CO2 levels are expected to change the relative concentrations of particulate and                                   |
| dissolved metals, due to the differential effects of high CO2 on trace metal solubility, speciation, adsorption and                                       |
| toxicity, as well as on the growth of different phytoplankton taxa, and their elemental trace metal composition. These                                    |
| processes will also be mediated by the presence of strong organic ligands in areas where particulate Fe inputs are                                        |
| important, since the effectiveness of some natural chelators such as siderophores, in dissolving Fe from oxyhydroxides                                    |
| and/or by enhancing the photoinduced redox cycle of Fe, will be increased, This study demonstrates the utility and                                        |
| robustness of combining trace metal analyses of particles in a controlled mesocosm experiment with manipulations of                                       |
| CO2 and Fe concentrations using natural assemblages of marine phytoplankton in order to understand future ocean                                           |
|                                                                                                                                                           |

41 phytoplankton

| 72  | 1. Introduction                                                                                                               |    |
|-----|-------------------------------------------------------------------------------------------------------------------------------|----|
| 73  | Marine phytoplankton contribute half of the world's total primary productivity, sustaining marine food webs and               |    |
| 74  | driving the biogeochemical cycles of carbon and nutrients (Field et al., 1998). Annually, phytoplankton incorporate           |    |
| 75  | approximately 45 to 50 billion metric tons of inorganic carbon (Field et al., 1998), removing a quarter of the CO2            |    |
| 76  | emitted to the atmosphere by anthropogenic activities (Canadell et al., 2007). Yet, the atmospheric CO2 concentration         |    |
| 77  | has increased by 40 % since pre-industrial times as a result of anthropogenic CO2 emissions, producing rapid changes          |    |
| 78  | in the global climate system (Stocker et al., 2013). The dissolution of anthropogenic CO2 in seawater, causes shifts in       |    |
| 79  | the carbonate chemical speciation, and leads to ocean acidification (OA). Marine ecosystems are sensitive to changes in       | l  |
| 80  | pH because pH strongly affects chemical and physiological reactions (Hoffman et al., 2012). Increased CO2 in seawater         |    |
| 81  | may enhance or diminish phytoplankton productivity (Mackey et al., 2015), decrease the CaCO3 production in most               |    |
| 82  | planktonic calcifiers (Riebesell and Tortell 2011), and/or inhibit organic nitrogen and phosphorus acquisition (Hutchins      |    |
| 83  | et al., 2009). Thus, the biogeochemical cycling of nutrients is predicted to be highly affected by OA (Hutchins et al.        |    |
| 84  | 2009), as well as the distribution and speciation of trace metals in the ocean (Millero et al., 2009).                        |    |
| 85  | A                                                                                                                             |    |
| 86  | Trace metals, including Fe, Zn, Mn, Cu, Co and Mo, are essential for biological functions (e.g. photosynthesis,               |    |
| 87  | respiration and macronutrient assimilation), and Cd can supplement these functions. Trace metals availability can             |    |
| 88  | influence phytoplankton growth and community structure (Morel and Price, 2003). In turn, plankton control the                 |    |
| 89  | distribution, chemical speciation, and cycling of trace metals in the sea (Sunda, 2012), by, for example, releasing           |    |
| 90  | organic compounds that dominate the coordination chemistry of metals, internalizing trace elements into the cells, and        |    |
| 91  | reducing and/or oxidizing metals at the cell surface. The chemistry of redox speciation of active trace metals is highly      |    |
| 92  | dependent on pH. For instance, Fe occurs in two main redox states in the environment: oxidized ferric Fe (Fe (III)),          |    |
| 93  | which is poorly soluble at circumneutral pH; and reduced ferrous Fe (Fe (II)), which is easily soluble and therefore          |    |
| 94  | more bioavailable. Fe speciation and bio- availability are dynamically controlled by the prevalent changing redox             |    |
| 95  | conditions. Also, as the ocean becomes more acidic, reduction of Cu (II) will increase, as the ionic form of Cu (II) is       |    |
| 96  | reduced to Cu (I) (Millero et al., 2009). The effect of higher concentrations of Cu (I) in surface waters on biological       |    |
| 97  | systems is not well known. Therefore, while the effects of OA on inorganic metal speciation will be more pronounced           |    |
| 98  | for metals that form strong complexes with carbonates (e.g. copper) or hydroxides (e.g. iron and aluminium), those that       |    |
| 99  | form stable complexes with chlorides (e.g. cadmium) will not be greatly affected. pH mediated changes in                      |    |
| 100 | concentrations and/or speciation could possibly enhance trace metals limitation and/or toxicity to marine plankton            |    |
| 101 | (Millero et al., 2009).                                                                                                       |    |
| 102 | A                                                                                                                             | •  |
| 103 | Iron is crucial for phytoplankton growth because of its involvement in many essential physiological processes, such as        |    |
| 104 | photosynthesis, respiration, and nitrate assimilation (Behrenfeld and Milligan, 2013). The decrease in seawater pH in         |    |
| 105 | response to OA may increase Fe solubility (Millero et al., 2009), but it may also result in unchanged or lower Fe             |    |
| 106 | bioavailability, depending of the nature of the strong organic Fe ligands (Shi et al., 2010). Consequently, changes in        |    |
| 107 | iron bioavailability due to ocean acidification can affect positively or negatively ocean productivity and $\mathrm{CO}_2$    |    |
| 108 | drawdown. Copper is an essential micronutrient but may be toxic at high concentrations (Semeniuk et al., 2016). An            |    |
| 109 | increase in free cupric ion concentrations in coastal areas due to ocean acidification (Millero et al., 2009) could result in | i. |
| 110 | negative effects on phytoplankton. From the open-ocean to coastal areas, the concentration of metals differ, as well as       | 1  |
| 111 | the trace metal requirements of phytoplankton (Sunda and Huntsman, 1995a), and their tolerance to metal toxicity.             |    |
| 112 | Accordingly, changes in pH may promote an increase in Cu toxicity in coastal phytoplankton, or enhance Fe limitation          | 1  |
| 113 | in the open ocean. Given that are essential for phytoplankton productivity, and that are actively                             |    |

**(Con formato: Inglés (británico)**

[revised manuscript text omitted]

|-------------------------------------------------------------------------------------------------------------------------------------------------------------------------------------------------------------------------------------------------------------------------------|
Después: 12 pto, Agregar espacio entre párrafos
del mismo estilo, No ajustar espacio entre texto
latino y asiático, No ajustar espacio entre texto
asiático y números, Punto de tabulación: No en
11.75 cm |
| (Flimingdo)                                                                                                                                                                                                                                                                   |

[revised manuscript text omitted]

|-------------------------------------------------|

|------------------------------------------------|
|                                                |

using a high-resolution inductively coupled plasma-mass spectrometer (ICP-MS, Element XR, Thermo Scientific) and
the described instrumental settings (Table S1). Filter blanks were collected and subjected to the same storage, digestion,
dilution, and analysis processes, and these blank values were subtracted from sample measurements. Particulate
samples for ICPMS analysis were processed in a trace metal-clean laboratory under a trace metal-clean laminar flow
fume hood.

**235 2.3.3 The effect of oxalate-EDTA wash on particulate trace metal concentrations**

236 To better estimate the biogenic fraction of the particulate metals, the filters were washed with an oxalate-EDTA 237 solution, which removes extracellular metals and oxyhydroxides (Tovar-Sanchez et al., 2003; Tang and Morel, 2006). 238 In our study, the oxalate wash significantly decreased the concentration of all particulate metals, with the exception of 239 Al and Ti (Tables S2 and S3), as observed by Rauschenberg and Twining (2015). The quantity of metal remaining after 240 the oxalate wash (i.e. biogenic fraction) varied among elements (Tables S2 and S3), In general, the concentrations of Fe 241 and Co in the particles were decreased the least by the oxalate wash by  $\sim 25\%$  while Mo and Pb concentrations were 242 decreased the most by  $\sim 70\%$  The concentrations of particulate Cu, Zn, Cd and Mn were reduced by 50% by the oxalate 243 wash. As shown previously (Sanudo-Wilhelmy et al. 2004), the oxalate reagent also removed extracellular P (by ~20%, 244 Table S2 & S3), Compared to Rauschenberg and Twining (2015), the estimates of the biogenic fraction, after the 245 oxalate wash, were in agreement for Co, Cu and P, and lower for Fe, Mn, Zn and Cd concentrations.

247 However, the efficacy of the oxalate wash to dissolve Fe, and other metals, from lithogenic particles is not well 248 constrained (Frew et al. 2006, Rauschenberg and Twining, 2015, King et al., 2012). Therefore, the results obtained after 249 the oxalate-EDTA wash should be interpreted with caution because we do not know whether the removed metal 250 fraction is a) only lithogenic; b) mainly lithogenic but some biogenic fraction is also removed, or c) whether metals 251 absorbed onto particles are equally labile to the wash on biogenic and lithogenic particles. Given that many of the 252 trends we observed were identical for the oxalate-EDTA washed and non-washed particles [i.e., higher Me 253 concentrations in the LC+DFB treatments (Table S2 & S3) and positive correlations between phytoplankton biomass 254 and Me concentrations [Lorenzo-Garrido 2016]], below we present and discuss only the non-oxalate wash results.

**256 2.4 Statistical analyses**

Data were checked for normality (by Shapiro-Wilks' test), homoscedasticity (by Levene's test) and sphericity (by
 Mauchly's test). All data met the requirements to perform parametric tests. Statistical significance of treatment effects
 was carried out using Split-Plot ANOVA followed by post-hoc Sidak and Bonferroni tests (considering P < 0.05 as</li>
 significant). All analyses were performed using the General Linear Model (GLM) procedure. The correlation between
 variables was analysed by Pearson's product-moment multiple comparisons (considering P < 0.05 as significant).</li>
 Statistical analyses were carried out using SPSS v22 (IBM statistics) and Sigmaplot 12 (Systat Software, Chicago,
 USA).

| 2 | 6 | 4 |
|---|---|---|
| Г |   |   |

246

255

265

|-------------------|-------------------------------------------------------------|
| //                | Movido hacia abajo[2]: Statistical analyses                 |
| ()                | Movido hacia abajo[3]: 3. Results ¶                         |
| (                 | Movido hacia abajo[4]: This bloom was not observed          |
|                   | Movido hacia abajo[5]: ¶                                    |
| (                 | Movido hacia arriba[1]: huxleyi bloom were ranked as:       |
|                   | Movido (inserción)[2]                                       |

| 676        | 3. Results                                                                                                                                                                                                                                          | Movido (inserción)[3]                           |
|------------|------------------------------------------------------------------------------------------------------------------------------------------------------------------------------------------------------------------------------------------------------------|-------------------------------------------------|
| 677        | 3.1 Biological and chemical characteristics during the bloom                                                                                                                                                                                               |
| 6/8        | Plankton community dynamics and their response to the applied treatments in the mesocosms are described in detail by                                                                                                                                       | Con formato: Fuente: 12 pto, ingles (britanico) |
| 6/9        | Segovia et al. (2017). Briefly, at the beginning of the experiment (days 1-10) a bloom of large chain-forming diatoms                                                                                                                               |                                                 |
| 680        | was observed, which declined by day 7 (Figure 2). This diatom bloom was associated with a sharp decrease in nitrate                                                                                                                                        |                                                 |
| (92        | and silicic acid concentrations (Figure S1-supplemental material). Picoeukaryotes, dominated the phytoplankton                                                                                                                                             |                                                 |
| 682        | community on day 8 (Figure 2). During the first 10 days of the experiment, there were no significant differences in the                                                                                                                                    |                                                 |
| 683        | chemical variables measured between the treatments (Figures 1 and S1). On day /, half of the mesocosms were                                                                                                                                                |                                                 |
| 084
695 | amended by adding DFB (+DFB treatments). Between day / and 1/, an increase in dFe was observed in all treatments,                                                                                                                                          |                                                 |
| 685        | except in the control (Figure 1). This increase in dFe was sustained for the entire experiment in the DFB treatments                                                                                                                                       |                                                 |
| 080        | (Figure 1). Dissolved Cu concentrations were not affected by the different treatments (Figure 1). After day 10, a                                                                                                                                          |                                                 |
| 68/        | massive bloom of the coccolithophore Emiliania huxleyi developed under LC +DFB condition (Figure 2), out-                                                                                                                                           |                                                 |
| 688        | competing the rest of the plankton groups (Figure 2) This bloom was not observed either in the control treatment (LC-                                                                                                                                      |
Movido (inserción)[4]                       |
| 689        | DFB) or in the HC treatments, although E. huxleyi was still the most abundant species in all treatments; with the                                                                                                                                   | Con formato: Ingles (britanico)                 |
| 090
601 | exception of the HC-DFB treatment (Figure 2).                                                                                                                                                                                                              |                                                 |
| 691        |                                                                                                                                                                                                                                                            |                                                 |
| 692        | 5,2 Particulate metal concentrations during the mesocosm experiment                                                                                                                                                                                        |
| 693        | The pMe concentrations (nM, mean of all treatments and dates) during the experiment were highest for Al, Fe and Zn,
A = A = A = A = A = A = A = A = A = A =                                                                                             |                                                 |
| 605        | and lowest for Cd, following unit trend: Al $\approx$ Fe $\approx$ 2n $\approx$ 11 $\geq$ Cd $\approx$ Mn $\geq$ Mo $\approx$ Fb $\geq$ Cd $\geq$ Cd (Figure 5, Table S2).                                                                                 |                                                 |
| 606        | Significant changes over time were observed for all particulate trace metal concentrations (re, Cu, Co, Zn, Cd, Mn, Md                                                                                                                                     |                                                 |
| 607        | and Pb), except for 11 and AI (Figure 5, 1 able 1). The only metal that showed a significant time-dependent decrease in                                                                                                                                    |                                                 |
| 697        | its particulate concentration was Fe (Figure 3, Table 1). In general, the treatments with the highest particulate metals                                                                                                                                   |                                                 |
| 600        | concentrations also exhibited the highest particulate P, except for AI, 11, Fe, and PO (Figure 5, Table S2). On days 12                                                                                                                                    |                                                 |
| 700        | and 17, the nights particulate metals concentrations were observed in the LC+DFB treatment, while on day 21, they                                                                                                                                          |                                                 |
| 700        | were observed in both LC treatments (Figure 3, Table S2).                                                                                                                                                                                                  |                                                 |
| 701        | 2.2 The effects of increased CO, and the DEP addition on neutinelate metal concentrations                                                                                                                                                                  |                                                 |
| 702        | 5.5 The effects of increased CO2 and the DFB addition on particulate metal concentrations                                                                                                                                                                  |                                                 |
| 704        | (Tables Land S2). Similarly, the addition of DEP did not directly influence particulate concentrations of Eq. but high                                                                                                                                     |                                                 |
| 705        | (Tables Taild 32). Similarly, the addition of DFB and not an edity influence particulate concentrations of Fe, but high                                                                                                                                    |                                                 |
| 705        | else inversely affected by CO 2 but only in the presence of DEP (CO 2 ) and CO 2 y DEP affect. Tables 1 and S2, Figure 5).                                                                                                |                                                 |
| 707        | <ol> <li>All other elements (P. Co. 7n. Mn and Mo) exhibited significant effects by CO2 and by DFR, but there was also a</li> </ol>                                                                                                             | Conformato: Fuanto: Sin Nagrita, Inglás         |
| 708        | 5). All other prements (1, Co, Zii, will and work control significant effects by Co 2 and by Dr D, but increases and a significant interaction between these two factors (Table 1, S2). This indicates that for example, particulate Mn, Zn, Mc |
(británico)                                 |
| 709        | Co. and P concentrations were significantly decreased by high CO 2 but only in the +DEB treatments (Figure 3 Table 1                                                                                                                            |                                                 |
| 710        | S2) Similarly the addition of DER significantly increased pZn and pMn, but only at ambient CO 2 levels (Figure 3                                                                                                                                |                                                 |
| 711        | Table 1 S2)                                                                                                                                                                                                                                                |                                                 |
| 712        | 1000 1,027                                                                                                                                                                                                                                          |                                                 |
| 713        | 3.4 Phospharous-normalized metal ratios in particles collected from the mesocosms and the effects of increased                                                                                                                                             |                                                 |
| 714        | $CO_2$ and the DFR addition on these ratios                                                                                                                                                                                                                |                                                 |
| 715        | The P-normalized metal ratios (Figure 4 and means in Table 2) were highest for $\Delta 1$ and Fe (mean: $70 \pm 38$ mmol $\Delta 1$ .                                                                                                                      |                                                 |
| 716        | mol P and $39 \pm 34$ mmol Fe mol P) and lowest for Cd and Co (mean 0.02 ± 0.01 mmol Cd; mol P, and 0.07 ± 0.02                                                                                                                                            |                                                 |
| 717        | mol C 0 mol P) Iron P and Ti P were not significantly affected by increased CO 0 and/or the DER addition, but                                                                                                                        |                                                 |
| 718        | showed a significant decrease over time (Table 3). The P_normalized Cu, Co and Zn ratios changed significantly over                                                                                                                                        |                                                 |
| /10        | snowed a signmean decrease over unit (rable 3). The r-normalized Cu, Co and Zir ratios enanged significantly over                                                                                                                                          |                                                 |

[revised manuscript text omitted]

Movido (inserción)[6] Con formato: Inglés (británico)

|------------------------------------------------------------------------------------------------------------------------------------------------------------------------------------------------------------------------------------------------------------------------------------------------------------------------------|
|                                                                                                                                                                                                                                                                                                                              |
| Movido hacia abajo[7]: huxleyi biomass was observed.                                                                                                                                                                                                                                                                         |
| Movido hacia abajo[7]: huxleyi biomass was observed.
| Movido hacia abajo[7]: huxleyi biomass was observed.
| Movido hacia abajo[7]: huxleyi biomass was observed.
| Movido hacia abajo[7]: huxleyi biomass was observed.
| Movido hacia abajo[7]: huxleyi biomass was observed.
| Movido hacia abajo[7]: huxleyi biomass was observed.
| Movido hacia abajo[7]: huxleyi biomass was observed.
| Movido hacia abajo[7]: huxleyi biomass was observed.
| Movido hacia abajo[7]: huxleyi biomass was observed.

| but         | their concentrations were lower than those in LC+DFB. At the end of the experiment, the concentrations of these                |
|-------------|--------------------------------------------------------------------------------------------------------------------------------|
| biog        | enic metals were, in general, comparable in both HC treatments, and lower than those in the LC treatments (Figure              |
| 3, T | able S3). Therefore, high CO2 had a tendency to decrease particulate metal concentrations, especially on day 21.               |
| Giv         | en the strong correlation between concentrations of these particulate bioactive metals and phytoplankton biomass,              |
| the         | ower particulate concentrations in high CO2 were mainly due to low phytoplankton biomass                                       |
|             |                                                                                                                                |
| Part        | iculate Zn concentrations were especially high in the LC+DFB treatment (Figure 3), where the highest E_huxleyi                 |
| bior        | nass was observed. Emiliania huxleyi is well known for its high Zn cellular requirements (~ 1-10 for E. huxleyi vs.            |
| 1-4         | mmol Zn: mol P for other phytoplankton; Sunda and Hunstman 1995, Sunda 2013). But, the Zn: P ratios in the                     |
| LC-         | DFB treatment (range 45-69 mmol Zn: mol P; Figure 4, Table S2), as well as in all the other treatment (range 16-               |
| 34 r        | mol Zn: mol P; Figure 4, Table S2) were significantly higher than these published ratios. This could be explained              |
| by,         | he adsorption of these metals to the outside of the cells, and/or anthropogenic inputs of Zn into the fjord. The Zn:P          |
| ratio       | is in the samples washed with the oxalate-EDTA were still high (range 28-57 for LC+DFB and 16-33 mmol Zn:                      |
| mol         | P in all other treatments, Table S3), thus adsorption might have not been significant. We hypothesize that                     |
| anth        | ropogenic aerosols which are rich in anthropogenic particulate metals, such as Zn and Cu (Perry et al. 1999; Narita            |
| et al       | . 1999), and have high percentage of Zn and Cu dissolution (ref.), might be the source of these high Zn                        |
| con         | centrations and ratios in the particles.                                                                                       |
|             |                                                                                                                                |
| Sim         | ilarly, the Cu:P ratios in the collected particles were relatively elevated $(1.4 \pm 0.8 \text{ mmol Cu: mol P})$ compared to |
| thos        | e of other phytoplankton, including E. huxleyi (Table 2). The dissolved (7.7±0.41 nM Cu, Figure 1) and particulate      |
| Cu o | concentrations (0.35±0.25 nM, Table S2) in our experiment were high, and similar to those previously measured in               |
| this        | fjord (Muller et al., 2005). Rain events (or wet deposition of anthropogenic aerosols), in this fjord result in high           |
| diss        | olved Cu and the active production of strong organic ligands by Synechococcus-to lower the free Cu                             |
| con         | centrations (Muller et al., 2005). Therefore, high Cu might be a general condition in this fjord, and indigenous               |
| plar        | kton might have developed physiological mechanisms to deal with high Cu, such as the production of organic                     |

plankton might have developed physiological mechanisms to deal with high Cu, such as the production of organic
ligands to prevent uptake (Vraspir and Butler, 2009), or of heavy-metal-binding peptides (phytochelatins) to lower Cu
toxicity inside the cell (Ahner and Morel, 1995; Ahner et al., 1995; Knauer et al., 1998). Since we measured high
particulate Cu, and Cu:P in our experiment, *Ex huxleyi* might have been relying mainly on phytochelatins to buffer high
intracellular Cu (Ahner et al., 2002).

861

\*

862 The Cd:P ratios (average  $0.024 \pm 0.01$  mmol Cd:mol P, Figure 4 and 6) were significantly lower than those in 863 phytoplankton and E. huxleyi, (0.36 mmol Cd:mol P, Figure 4 and 6), This was surprising, because Cd quotas are 864 normally higher in coccolithophores than in diatoms and chlorophytes (Sunda and Huntsman, 2000; Ho et al., 2003). 865 High Cd quotas in coccolithophores have been suggested to result from accidental uptake through Ca transporters and 866 channels (Ho et al., 2009). The low Cd quotas here may be explained by the antagonistic interaction between Mn and 867 Cd or Zn and Cd under high Mn and Zn, respectively (Sunda and Huntsman, 1998, 2000; Cullen and Sherrell, 2005). 868 Since high Zn:P ratios were common in this study ( $34.02 \pm 18.05 \text{ mmol } \text{Zn:mol } \text{P}$ , Figure 4 and 6), we hypothesize that 869 high Zn levels antagonistically interacted with Cd, resulting in low Cd:P ratios in the particles.

871 4.4 The effects of increased CO2 and the DFB addition on particulate metal concentrations and P-normalized

870

872 ratios

Iron enrichment is common in coastal waters, due to sediment resuspension, rivers input, aeolian deposition and mixing

or upwelling of deep water. Indeed, Fe was the essential metal with the highest particulate concentrations in our study

|---|----------------------------------------------------------|----------|---------------|
|   | Movido (inserción)[7]                                    |          |               |
|   | Eliminado: Emiliania huxleyi is well known for its       | hig      | h Z 81 |
|   | Con formato                                              |          | [[ 2]         |
111e | [CC]          |
for  | [56]          |
|   | Con formato                                              | 101      | [4 6 H        |
| / | Con formatio                                             |          | [59]          |
|   | Conformato                                               |          | [60]          |
|   | Eliminado: The high Zn:P ratios in this study indica     | ite.     | h ise] |
| / | Movido hacia arriba[6]: ¶                                |          |               |
|   | (                                                        |          |               |

[revised manuscript text omitted]

1043 5. Concluding remarks

| Fliminado: was Fe (                                                                                                                                                                      |
|------------------------------------------------------------------------------------------------------------------------------------------------------------------------------------------|
| correlations with Al and without correlation with                                                                                                                                        |
| phytoplankton biomass (Figure 1, Table 5), indicating                                                                                                                                    |
| and that of the crustal ratio (Figure 1) also supports this finding. Indeed, the Fe P ratios were significantly higher than                                                              |
| those of indigenous plankton assemblages                                                                                                                                                 |
| Movido hacia abajo[8]: 2009).                                                                                                                                                            |
| Movido (inserción)[8]                                                                                                                                                                    |
| com comator marco (oritanico)                                                                                                                                                            |

14 The results presented here show that in the fjord where we carried out the present experiment, particulate Fe was dominated by lithogenic material, and was significantly decreased by future predicted CO2 concentrations (HC, 900 16 µatm) and DFB addition. This condition may well be comparable to most coastal ecosystems in the future ocean. 17 Indeed, high CO2 and/or DFB promoted the dissolution of particulate Fe, and the presence of this strong organic 18 complex helped maintaining high dissolved Fe. Under control conditions at present CO2 concentration (LC, 380 µatm) 19 and no DFB amendment, the globally important coccolithophore Emiliania huxleyi was experiencing Fe limitation 1120 (Segovia et al. 2017). The shift between particulate and dissolved Fe promoted a massive bloom of E. huxleyi in the 1121 treatments with ambient CO2 due to increased Fe bioavailability (for further details on Fe-bioavailability in E. huxleyi 22 please see Segovia et a. 2017). Moreover, the negative effects of high CO2 were mitigated by enhanced dFe. During the 23 mentioned bloom, the concentrations of particulate metals with a strong biogenic component (Cu, Co, Zn, Cd, Mn, and 124 Mo) were a) highly dynamic, b) positively correlated with plankton biomass, and c) influenced by growth requirements. 1125 Furthermore, high CO2 decreased the Me:P ratios of Co, Zn and Mn, mainly due to low phytoplankton biomass, while 1126 increased the Cu:P ratios. In contrast DFB had no effects on these ratios. According to our results, high CO2 may 27 decrease particulate Fe and increase dissolved Fe, but high concentrations of dissolved Fe will only be maintained by 128 the presence of strong organic ligands. The decrease in particulate Fe may affect the sinking flux of other metals 129 associated with terrestrial material/dust in open ocean settings. Furthermore, ocean acidification will decrease E. 1130 huxleyi abundance, and as a result, the concentration of many biogenic particulate metals. Moreover, the Me:P ratios of 131 metals that are predominately present in an ionic free form in seawater (e.g. Co, Zn and Mn) will likely decrease or stay 32 constant. However, the high pZn observed will possibly be the result of anthropogenic aerosols, and the responsible for 33 the low pCd registered, most likely due to the antagonistic interaction between Zn and Cd. In contrast, high CO2 is 134 predicated to shift the speciation of dissolved metals associated with carbonates, such as Cu, increasing their 1135 bioavailability, and resulting in higher Me:P ratios. We suggest that high Cu might be putative in this fjord, and 1136 autochthonous plankton might be adapted to cope with high Cu levels by developing specific physiological 37 mechanisms. Future predicted high CO2 levels are expected to change the relative concentrations of particulate and 138 dissolved metals, due to the differential effects of high CO2 on trace metal solubility, speciation, adsorption and 139 toxicity, as well as on the growth of different phytoplankton taxa, and their elemental trace metal composition. In the 1140 future ocean, this will have great implications in the carbon cycle and the biological pump, consequently affecting the 41 physic, chemical and biological aspects, i.e. marine systems dynamics. 42

**143 Acknowledgments**

1144This work was funded by CTM/MAR 2010-17216 (PHYTOSTRESS) research grant from the Spanish Ministry for1145Science and Innovation (Spain) to MS, and by NSERC grants (Canada) to MTM and JTC. MRL was funded by a FPU1146grant from the Ministry for Education (Spain) and by fellowships associated to the mentioned above research grants to1147carry out a short-stay at MTM and JTC laboratories to analyze, dissolved and particulate metals. We thank all the1148participants of the PHYTOSTRESS experiment for their collaboration, and the MBS (Espegrend, Norway) staff for1149logistic support during the experiment.

| Página 6: [1] Con formato                     | María Segovia                    | 11/6/19 19:41:00      |
|-----------------------------------------------|----------------------------------|-----------------------|
| Inglés (británico)                            |                                  |                       |
| Página 6: [2] Con formato                     | María Segovia                    | 11/6/19 19:41:00      |
| Ninguno, Agregar espacio entre p
huérfanas | árrafos del mismo estilo, Contro | ol de líneas viudas y |
| Página 6: [3] Con formato                     | María Segovia                    | 11/6/19 19:41:00      |
| Color de fuente: Texto 1, Inglés (l           | oritánico)                       |                       |
| Página 6: [4] Con formato                     | María Segovia                    | 11/6/19 19:41:00      |
| Inglés (británico)                            |                                  |                       |
| Página 6: [5] Con formato                     | María Segovia                    | 11/6/19 19:41:00      |
| Fuente: 12 pto, Inglés (británico)            |                                  |                       |
| Página 6: [6] Eliminado                       | María Segovia                    | 11/6/19 19:41:00      |
| ۷                                             |                                  |                       |
| Página 6: [7] Con formato                     | María Segovia                    | 11/6/19 19:41:00      |
| Fuente: Negrita, Color de fuente:             | Texto 1, Inglés (británico)      |                       |
| Página 6: [8] Eliminado                       | María Segovia                    | 11/6/19 19:41:00      |
| ×                                             |                                  |                       |
| Pégina 6, [0] Can formata                     | María Sagavia                    | 11/6/10 10:41:00      |
| Inglés (británico)                            | Maria Seguvia                    | 11/6/19 19:41:00      |
| Página 6: [10] Eliminado                      | Ματία δοσογία                    | 11/6/10 10:41:00      |
|                                               |                                  | 11/0/19 19.41.00      |
| · · · · · · · · · · · · · · · · · · ·         |                                  |                       |
| Página 6: [11] Con formato                    | María Segovia                    | 11/6/19 19:41:00      |
| Color de fuente: Texto 1, Inglés (l           | oritánico)                       |                       |
| Página 6: [12] Con formato                    | María Segovia                    | 11/6/19 19:41:00      |
| Color de fuente: Texto 1, Inglés (l           | oritánico)                       |                       |
| Página 6: [13] Eliminado                      | María Segovia                    | 11/6/19 19:41:00      |
| Página 6: [14] Con formato                    | María Segovia                    | 11/6/19 19:41:00      |
| Color de fuente: Texto 1, Inglés (l           | oritánico)                       |                       |
| Página 6: [15] Eliminado                      | María Segovia                    | 11/6/19 19:41:00      |
| ۷                                             |                                  |                       |
| Página 6: [16] Con formato                    | María Segovia                    | 11/6/19 19:41:00      |
| Color de fuente: Texto 1, Inglés (l           | pritánico)                       |                       |
| Página 6: [17] Eliminado                      | María Segovia                    | 11/6/19 19:41:00      |
| Página 6: [18] Con formato                    | María Segovia                    | 11/6/19 19:41:00      |
| Inglés (británico)                            | mana segona                      | 11/0/13 13.41.00      |
| Página 6: [18] Con formato                    | María Segovia                    | 11/6/19 19:41:00      |
|                                               | maria Seguria                    | 11/0/13 13.41.00      |

Inglés (británico)

| Página 6: [19] Eliminado                | María Segovia | 11/6/19 19:41:00 |
|-----------------------------------------|---------------|------------------|
| ▼                                       |               |                  |
| Página 6: [20] Con formato              | María Segovia | 11/6/19 19:41:00 |
| Inglés (británico)                      |               |                  |
| Página 6: [21] Eliminado                | María Segovia | 11/6/19 19:41:00 |
| ×                                       | -             |                  |
| Página 6: [22] Con formato              | María Sogovia | 11/6/10 10:41:00 |
| Color de fuente: Texto 1 Inglés (brité  |               | 11/0/19 19.41.00 |
| Página 6: [23] Con formato              | María Segovia | 11/6/19 19:41:00 |
| Color de fuente: Texto 1. Inglés (brité | inico)        | 11/0/19 19.41.00 |
| Página 6: [23] Con formato              | María Segovia | 11/6/19 19:41:00 |
| Color de fuente: Texto 1. Inglés (britá | inico)        | 11/0/15 15.41.00 |
| Página 6: [24] Con formato              | María Segovia | 11/6/19 19:41:00 |
| Color de fuente: Texto 1, Inglés (britá | inico)        |                  |
| Página 6: [25] Con formato              | María Segovia | 11/6/19 19:41:00 |
| Color de fuente: Texto 1, Inglés (britá | inico)        |                  |
| Página 6: [26] Con formato              | María Segovia | 11/6/19 19:41:00 |
| Color de fuente: Texto 1, Inglés (britá | inico)        |                  |
| Página 6: [27] Con formato              | María Segovia | 11/6/19 19:41:00 |
| Color de fuente: Texto 1, Inglés (britá | nico)         |                  |
| Página 6: [28] Con formato              | María Segovia | 11/6/19 19:41:00 |
| Color de fuente: Texto 1, Inglés (britá | inico)        |                  |
| Página 6: [29] Con formato              | María Segovia | 11/6/19 19:41:00 |
| Inglés (británico)                      |               |                  |
| Página 6: [30] Con formato              | María Segovia | 11/6/19 19:41:00 |
| Inglés (británico)                      |               |                  |
| Página 6: [31] Con formato              | María Segovia | 11/6/19 19:41:00 |
| Inglés (británico)                      |               |                  |
| Página 6: [31] Con formato              | María Segovia | 11/6/19 19:41:00 |
| Inglés (británico)                      |               |                  |
| Página 6: [32] Con formato              | María Segovia | 11/6/19 19:41:00 |
| Color de fuente: Texto 1, Inglés (britá | inico)        |                  |
| Página 6: [32] Con formato              | María Segovia | 11/6/19 19:41:00 |
| Color de fuente: Texto 1, Inglés (brité | inico)        |                  |
| Página 6: [33] Eliminado                | María Segovia | 11/6/19 19:41:00 |

| Página 6: [34] Con formato                        | María Segovia           | 11/6/19 19:41:00                  |
|---------------------------------------------------|-------------------------|-----------------------------------|
| Inglés (británico)                                |                         |                                   |
| Página 6: [35] Con formato                        | María Segovia           | 11/6/19 19:41:00                  |
| Inglés (británico)                                |                         |                                   |
| Página 6: [36] Con formato                        | María Segovia           | 11/6/19 19:41:00                  |
| Nivel 1, No agregar espacio entre pá
huérfanas | rrafos del mismo estilo | , Sin control de líneas viudas ni |
| Página 6: [37] Con formato                        | María Segovia           | 11/6/19 19:41:00                  |
| Inglés (británico)                                |                         |                                   |
| Página 10: [38] Con formato                       | María Segovia           | 11/6/19 19:41:00                  |
| Inglés (británico)                                |                         |                                   |
| Página 10: [39] Con formato                       | María Segovia           | 11/6/19 19:41:00                  |
| Inglés (británico)                                |                         |                                   |
| Página 10: [40] Con formato                       | María Segovia           | 11/6/19 19:41:00                  |
| Inglés (británico)                                |                         |                                   |
| Página 10: [41] Con formato                       | María Segovia           | 11/6/19 19:41:00                  |
| Inglés (británico)                                |                         |                                   |
| Página 10: [42] Con formato                       | María Segovia           | 11/6/19 19:41:00                  |
| Inglés (británico)                                |                         |                                   |
| Página 10: [43] Con formato                       | María Segovia           | 11/6/19 19:41:00                  |
| Inglés (británico)                                |                         |                                   |
| Página 10: [44] Con formato                       | María Segovia           | 11/6/19 19:41:00                  |
| Inglés (británico)                                |                         |                                   |
| Página 10: [45] Eliminado                         | María Segovia           | 11/6/19 19:41:00                  |
| ▼                                                 |                         |                                   |
| Página 10: [46] Con formato                       | María Segovia           | 11/6/19 19:41:00                  |
| Inglés (británico)                                |                         | · ·                               |
| Página 10: [47] Con formato                       | María Segovia           | 11/6/19 19:41:00                  |
| Inglés (británico)                                |                         |                                   |
| Página 10: [48] Eliminado                         | María Segovia           | 11/6/19 19:41:00                  |
| · · · · · ·                                       |                  | · ·                               |
|                                                   |                         |                                   |
| Pagina 10: [49] Con formato                       | Maria Segovia           | 11/6/19 19:41:00                  |
| ingles (britanico)                                |                         |                                   |
| Pagina 10: [50] Eliminado                         | Maria Segovia           | 11/6/19 19:41:00                  |
| V                                          |                         |                                   |
| Página 10: [51] Con formato                       | María Segovia           | 11/6/19 19:41:00                  |

Inglés (británico)

I

| ágina 10: [52] Con formato         | María Segovia                 | 11/6/19 19:41:00           |
|------------------------------------|-------------------------------|----------------------------|
| nglés (británico)                  |                               |                            |
| ágina 10: [53] Con formato         | María Segovia                 | 11/6/19 19:41:00           |
| nglés (británico)                  |                               |                            |
| ágina 10: [54] Eliminado           | María Segovia                 | 11/6/19 19:41:00           |
|                                    |                               |                            |
| ágina 10: [55] Con formato         | María Segovia                 | 11/6/19 19:41:00           |
| nglés (británico)                  |                               |                            |
| ágina 10: [56] Con formato         | María Segovia                 | 11/6/19 19:41:00           |
| nglés (británico)                  |                               |                            |
| ágina 10: [57] Eliminado           | María Segovia                 | 11/6/19 19:41:00           |
|                                    |                               |                            |
|                                    |                               |                            |
| ágina 10: [58] Con formato         | María Segovia                 | 11/6/19 19:41:00           |
| ngles (britanico)                  |                               |                            |
| ágina 10: [59] Eliminado           | María Segovia                 | 11/6/19 19:41:00           |
| ágina 10: [60] Con formato         | María Segovia                 | 11/6/19 19:41:00           |
| erecha: 0 cm, Agregar espacio es   | ntre párrafos del mismo estil | o, Punto de tabulación: No |
| n 11.75 cm                         |                               |                            |
| ágina 10: [61] Con formato         | María Segovia                 | 11/6/19 19:41:00           |
| nglés (británico)                  |                               |                            |
| ágina 10: [62] Con formato         | María Segovia                 | 11/6/19 19:41:00           |
| nglés (británico)                  |                               |                            |
| ágina 10: [63] Con formato         | María Segovia                 | 11/6/19 19:41:00           |
| nglés (británico)                  |                               |                            |
| ágina 10: [64] Con formato         | María Segovia                 | 11/6/19 19:41:00           |
| uente: Cursiva, Inglés (británico) |                               |                            |
| ágina 10: [65] Con formato         | María Segovia                 | 11/6/19 19:41:00           |
| nglés (británico)                  |                               |                            |
| ágina 10: [66] Eliminado           | María Segovia                 | 11/6/19 19:41:00           |
|                                    |                               |                            |
| ágina 10: [67] Con formato         | María Segovia                 | 11/6/19 19:41:00           |
| nglés (británico)                  |                               |                            |
| ágina 10: [68] Con formato         |                               |                            |
|                                    | María Segovia                 | 11/6/19 19:41:00           |
| nglés (británico)                  | María Segovia                 | 11/6/19 19:41:00           |

Derecha: 0 cm, Espacio Después: 12 pto, Sin control de líneas viudas ni huérfanas, No ajustar espacio entre texto latino y asiático, No ajustar espacio entre texto asiático y números, Punto de tabulación: No en 11.75 cm

| Página 10: [70] Con formato           | María Segovia     | 11/6/19 19:41:00 |  |
|---------------------------------------|-------------------|------------------|--|
| Inglés (británico)                    |                   |                  |  |
| Página 10: [71] Con formato           | María Segovia     | 11/6/19 19:41:00 |  |
| Inglés (británico)                    |                   |                  |  |
| Página 11: [72] Eliminado             | María Segovia     | 11/6/19 19:41:00 |  |
| v                                     |                   |                  |  |
|                                       |                   |                  |  |
| Página 11: [73] Eliminado             | María Segovia     | 11/6/19 19:41:00 |  |
| ×                                     |                   |                  |  |
|                                       |                   |                  |  |
| Página 11: [74] Eliminado             | María Segovia     | 11/6/19 19:41:00 |  |
| · · · · · · · · · · · · · · · · · · · |                   |                  |  |
|                                       |                   |                  |  |
| Página 11: [75] Con formato           | María Segovia     | 11/6/19 19:41:00 |  |
| Fuente de párrafo predeter Fuente     | e: 12 pto Español | 11/0/15 15.41.00 |  |
| Página 11: [76] Con formato           | María Sagovia     | 11/6/19 19:41:00 |  |
| Fuente de párrafo predeter Euente     | 2: 12 pto Español | 11/0/19 19.41.00 |  |
| Página 11: [77] Eliminado             | María Sagavia     | 11/6/10 10:41:00 |  |
|                                       | Maria Seguvia     | 11/0/19 19.41.00 |  |
| V                                     |                   |                  |  |
| Désine 11, [79] Can farmata           | María Caravia     | 11/6/10 10:41:00 |  |
| Fuente de nérrefe predeter Euente     | Maria Segovia     | 11/6/19 19:41:00 |  |
| Fuence de partato predeter., Fuence   |                   |                  |  |
| Página 11: [79] Eliminado             | Maria Segovia     | 11/6/19 19:41:00 |  |
| V                              |                   |                  |  |
|                                       |                   |                  |  |
| Página 11: [80] Con formato           | María Segovia     | 11/6/19 19:41:00 |  |
| Fuente: Sin Negrita, Inglés (britán   | ico)              |                  |  |
| Página 11: [81] Eliminado             | María Segovia     | 11/6/19 19:41:00 |  |
| ▼                                     |                   |                  |  |
|                                       |                   |                  |  |
| Página 11: [82] Eliminado             | María Segovia     | 11/6/19 19:41:00 |  |
| ▼                                     |                   |                  |  |
|                                       |                   |                  |  |
| Página 11: [83] Eliminado             | María Segovia     | 11/6/19 19:41:00 |  |
|                                       |                   |                  |  |

| Página 11: [84] Con formato           | María Segovia                    | 11/6/19 19:41:00  |
|---------------------------------------|----------------------------------|-------------------|
| Fuente: Times, Color de fuente: Color | personalizado(RGB(21;21;24)), Ir | nglés (británico) |
| Página 11: [85] Con formato           | María Segovia                    | 11/6/19 19:41:00  |
| Fuente: Times, Color de fuente: Color | personalizado(RGB(21;21;24)), Ir | nglés (británico) |
| Página 11: [86] Eliminado             | María Segovia                    | 11/6/19 19:41:00  |
|                                       |                                  |                   |

| Página 11: [87] Con formato       | María Segovia | 11/6/19 19:41:00 |
|-----------------------------------|---------------|------------------|
| Fuente: Times 12 nto Inglés (brit | ánico)        |                  |

Fuente: Times, 12 pto, Inglés (británico)

| Factor                | Al | Ti | Р  | Fe  | Cu | Co  | Zn  | Cd  | Mn  | Mo  | Pb |
|-----------------------|----|----|----|-----|----|-----|-----|-----|-----|-----|----|
| CO 2       | ns | ns | ** | *   | ns | **  | *** | *** | **  | *** | ns |
| DFB                   | ns | ns | *  | ns  | ns | *   | **  | ns  | *   | *   | ns |
| CO 2 x DFB | ns | *  | ** | ns  | ns | *   | **  | *   | **  | **  | ns |
| Time                  | ns | ns | ns | *** | *  | *** | *** | *** | *** | *** | ** |

534  $\frac{111110}{ns: not significant; * p < 0.05; ** p < 0.01; *** p < 0.001}$

 Table 2. The average metal ratios in the particles collected in this study (without oxalate wash) using the data reported in Table S2. The Pnormalized ratios (mmol : mol P, Figure 4) are compared to previous estimates in marine plankton samples and phytoplankton cultures (A).
 The Al-normalized ratios (mmol : mol Al) are compared to crustal ratios (B).

543 A)

| (mmol : mol P)        | Mn:P            | Fe:P      | Co:P            | Cu:P       | Zn:P        | Cd:P            | Mo:P         | Al:P   | Reference         |
|-----------------------|-----------------|-----------|-----------------|------------|-------------|-----------------|--------------|--------|-------------------|
|                       |                 |           |                 |            |             |                 |              |        |                   |
| Phytoplankton Lab     | 3.8             | 7.5       | 0.19            | 0.38       | 0.8         | 0.21            | 0.03         |        | Ho et al.
2003 |
| Marine Plankton Field | 0.68±0.54       | 5.1±1.6   | 0.15±0.06       | 0.41±0.16  | 2.1±0.88    |                 |              |        | Ho 2006           |
| E. huxleyi Lab        | 7.1±0.36        | 3.5±0.07  | $0.29 \pm 0.02$ | 0.07±0.013 | 0.38±0.002  | 0.36±0.01       | 0.022±0.0003 |        | Ho et al.
2003 |
| This study            | $1.65 \pm 0.41$ | 39.2±34.3 | $0.07 \pm 0.02$ | 1.41±0.55  | 34.02±18.05 | $0.02 \pm 0.01$ | 0.42±0.12    | 70±38  |                   |
| Crust ratio           | 510             | 29,738    | 13              | 25         | 32          | 0.05            | 0.46         | 89,972 | Taylor 1964       |

544

| 545 | B) |                 |       |         |         |         |         |         |         |         |          |
|-----|----|-----------------|-------|---------|---------|---------|---------|---------|---------|---------|----------|
|     |    | (mmol : mol Al) | Mn:Al | Fe:Al   | Co:Al   | Cu:Al   | Zn:Al   | Cd:Al   | Mo:Al   | Pb:Al   | Ti:Al    |
|     |    | Crustal ratio   | 5.7   | 331     | 0.14    | 0.27    | 0.35    | 0.001   | 0.005   | 0.02    | 39       |
|     |    | This study      | 35±28 | 506±342 | 1.5±1.2 | 26.5±15 | 795±865 | 0.5±0.4 | 8.6±6.5 | 4.9±3.9 | 119±47.6 |

546 547

| 548 | Table 3. Statistical analyses (Split-plot ANOVA) of the effects of CO2, DFB, and their interaction, as well as the effect of time, on the P- |
|-----|----------------------------------------------------------------------------------------------------------------------------------------------|
| 549 | normalized metal quotas (mmol: mol P, data in Figure 4, and Table S2) in particles collected from the different mesocosm treatments.         |
|     |                                                                                                                                              |

550

| Factor          | Fe:P | Cu:P | Co:P | Zn:P | Cd:P | Mn:P | Mo:P | Pb:P | Ti:P |
|-----------------|------|------|------|------|------|------|------|------|------|
| CO 2 | ns   |      | ***  | **   | ns   |      | ns   | ns   | ns   |
| DFB             | ns   |
| CO2 x DFB       | ns   |
| Time            | ***  | ***  | ***  | ***  | ns   | ns   | ns   | ns   | ***  |

551 552

553

÷

1

1

555 556 557

 Table 4. The relationship (Pearson correlations, p < 0.05) between particulate metals concentrations (nmol L-1, no oxalate wash, reported in Table S2) and the biomass ( $\mu$ gC L-1) of *Emiliania huxleyi* and total cells (phytoplankton and microzooplankton) collected from the different mesocosm treatments.

|             |             | Р     | Fe | Cu    | Co        | Zn        | Cd        | Mn        | Мо        | Pb | Ti |
|-------------|-------------|-------|----|-------|-----------|-----------|-----------|-----------|-----------|----|----|
|             |             |       |    |       |           |           |           |           |           |    |    |
| E. huxleyi  | Correlation | 0.622 |    | 0.614 | 0.764     | 0.747     | 0.010     | 0.686     | 0.926     |    |    |
|             | coefficient |       | ns | 0.014 | 0.730     | 0.747     | 0.818     | 0.686     | 0.825     | ns | ns |
|             | P-value     | 0.003 |    | 0.003 | 7.35.10.5 | 1.01.10-4 | 6.02.10-6 | 5.93-10-4 | 4.20.10-6 |    |    |
| Total cells | Correlation | 0.641 |    | 0.51  | 0.44      | 0.000     | 0.003     | 0.500     | 0.52      |    |    |
|             | coefficient |       | ns | 0.51  | 0.644     | 0.889     | 0.802     | 0.598     | 0.53      | ns | ns |
|             | P-value     | 0.002 |    | 0.02  | 1.62.10-3 | 7.03.10-8 | 1.23.10-5 | 4.18-10-3 | 1.35.10-2 |    |    |
|             |             |       |    |       |           |           |           |           |           |    |    |

... [2]

558 🏌

| Página 4: [2] Eliminado María Segovia 11/6/19 19:50:00 | Página 1: [1] Eliminado | María Segovia | 11/6/19 19:50:00 |
|--------------------------------------------------------|-------------------------|---------------|------------------|
|                                                        | Página 4: [2] Eliminado | María Segovia | 11/6/19 19:50:00 |

| 27
28
29 | Supplemental_Figures                                                                                                            | Con formato: Numeración: Continua |
|----------------|---------------------------------------------------------------------------------------------------------------------------------|-----------------------------------|
| 30             | Particulate trace metal dynamics in response to increased CO2 and                                                               |                                   |
| 31             | iron availability in a coastal mesocosm experiment                                                                              |                                   |
| 32             | M. Rosario Lorenzo 1 , María Segovia 1 , Jay T. Cullen 2 , and María T. Maldonado 3 |                                   |
| 33             | Correspondence to: María Segovia (segovia@uma.es) and María T. Maldonado (mmaldonado@eoas.ubc.ca)                               |                                   |
| 34
35       |                                                                                                                                 |                                   |

---

## Editor Decision (ED1)

Review comments for "Particulate trace metal dynamics in response to increased CO2 and iron availability in a coastal mesocosm experiment".

The authors have improved this manuscript with addressing my comments. My serious concern is mostly resolved. However, the authors did not resolve all of my concerns. I still find several points which need to be improved before publication.

I have read this version of manuscript carefully, and I still feel that explanation of statistically analysis for Table 1 is not kind for reader. Why author do not describe following information in the text or caption of Table 1?

"We used all the days because we performed a Split-Plot ANOVA (or mixed model) which integrates fixed factors (Co2 and Fe) and a repeated measures factor (time) by using the posthoc Bonferroni, saying that the statistical treatment was a split-plot ANOVA+Bonferroni, compulsory means that time was fully considered during the whole experimental period."

They have made the response only to reviewer's comments. This information is very important for reader's understanding that how the statistically analysis have done.

Line 125 "By day 17,······and/or the addition of DFB (Chen et al., 2004)." Increased dissolved Fe by adding DFB is bioavailable? Is DFB-Fe can be detected by CL-FIA which described in Segovia et al., 2017? Strong chelate like DFB prevent dissolved Fe detection measuring by resin preconcentrate-CL-FIA measurement system. It should be made clear that which chemical species do authors describe as for dissolved Fe in this study (Is DFB-bounded Fe included in this dissolved fraction, or not?). Also, it is necessary to clearly describe how do authors think about that how the availability of iron was changed by adding DFB. Some previous studies indicate that DFB-bounded Fe is not available, as authors described in the text. Is DFB-Fe available for E. Huxley? If so, please indicate a reference. Or, do adding DFB induce other chemical species of dissolved fraction? Author should describe this aspect clear, because this point is very important for this study.

Line 130-132, "Water samples from······onshore laboratory." This sentence should be moved in section 2.3.1. Authors should indicate manufacture and model information for the "vacuum pump".

Line 145, "for this very experiment.". What is "very".

Line 155, Authors should indicate the type of "Filters". Is this also AcroPac Supore, but membrane type?? Size??

Line 158, "without manipulation" should be changed to "without oxalate-EDTA wash".

Line 164, Authors should indicate material and volume of "centrifuge tubes"

Line 178, blank value should be appeared in the Supplemental material such as S-Table 1.

Line 215-216, "This diatom bloom was associated with a sharp decrease in nitrate and silicate acid concentration". I think this is not correct. Why nitrate decrease with diatom decreasing? Silicate have not sharp decreased during diatom decreasing.

Line 327, "Figure 4" should be changed "Figure 3".

429-430, "promoted a massive bloom of E huxley in the treatment with ambient CO2, due to increased dissolved Fe". It should be made clear that which chemical species do authors describe as for dissolved Fe in this study (Is DFB-bounded Fe included in this dissolved fraction, or not?). Some previous studies indicate that DFB-bounded Fe is not available, as authors described in the text. Is DFB-Fe available for E. Huxley? If so, please indicate reference. See comment above too.

Line 435-436, "The decrease in particulate Fe may⋯.. in open ocean setting.". Delete this sentence. This is not a conclusion from this study. No data from this study indicate this.

End of review.

---

## Author Response (AR2)

* * *
**Referee #1**

**The authors have done an acceptable job in addressing my comments as well as those of other reviewers. Therefore, I would recommend publication.**

**Some minor comments.**

**Lines 62-65. "For instance, Fe occurs…more bioavailable.": first, references are needed for this sentence; second, what described here actually cannot serve as examples for the dependence of trace metal chemistry on seawater pH.**

We removed the clause about Fe II being more bioavailable, and added that Fe II is more soluble but rapidly oxidize according to Millero et al 1987: *"Iron occurs in two main redox states in the environment: oxidized ferric Fe (Fe (III)), which is poorly soluble at circumneutral pH; and reduced ferrous Fe (Fe (II)), which is more soluble in natural seawater, but becomes rapidly oxidized (Millero et al. 1987) "*

**Line 84. ", and that are actively…" should be ", and that they are actively…"**

Corrected

**Line 85. the impact on, not in, the trace metal content…**

Corrected

**Line 97. "E. huxleyi"…please also check the rest of the manuscript**

Checked

**Line 100. 1) Delete "organic", as CaCO3 is not organic, 2) delete "carbon export", and 3) Hutchings, 2011 is not in the reference list.**

Done

**Line 104. What "realistic" refers to here?**

We have clarified this point, it now reads as: *"Specifically, mesocosm experiments allow perturbation studies with a high degree of realism compared to other experimental systems such as in the laboratory (high controlled conditions usually far from reality) or in situ in the ocean (where not all the interactions are contemplated) (Riebesell et al., 2010, Stewart et al., 2013, Riebesell and Gatusso, 2015)"*

**Line 123. Replace "iron" with Fe, and also check the rest of the manuscript.**
**Done**

**Line 127. Should be "resulting in…"**

Done

**Line 135. Please provide the vendor of HCl in the first place.**

It was said in Ln 200: Fisher chemicals. We have moved this to the line above.

**Line 252. The effects of…on, not in, the plankton community**

Done

**Line 393. E. huxleyi.**

Done, now Ln 403

**Lines 401-402. At low pH, CO3 concentration decreases, which should lead to a decrease in CuCO3.**

Ok

**Although the diatom bloom is not the primary focus of this study, I would suggest the authors briefly discuss how the change in CO2 may affect the biogenic fraction of particulate Co, Zn, and Cd at the beginning of the mesocosm experiment.**

We do not agree about talking about diatoms. The paper is already pretty complex. And we do not want to speculate on diatoms, they were all gone (by day 6-7) before trace metal dynamics started to be affected. This was an Ehux bloom and so this is the main focus.
* * *
**Referee #2**

**"Particulate trace metal dynamics in response to increased CO2 and iron availability in a coastal mesocosm experiment"**
**Lorenzo et al.**

**-       General comments**
**Authors have responded to most of my previous comments.**
**I have found significant improving of this manuscript, especially for presenting their results with adequate "Figures" which is more understandable for readers. However, I still found some issues for this manuscript which have to concern before publications. Please consider for following comments.**

**Since authors present their data with some "Figures", it became more clearly that parts of presentation in this manuscript (MS) are overlapped to previous published paper, Segovia et al. (2017). For instance, Figure 1a, 1b and 2 are totally same figures as reported by Segovia et al. (2017). Is it really possible to publish only with the statement as "Figure reproduced with permission from Segovia et al. Mar. Ecol. Prog. Ser. 2017" in the caption? It might be better to place in "supplementary Figures". I would like to ask about this aspect to Editor.**

The copyright is all right. We have transferred them to supplemental material and re-numbered all figures. The partition coefficients that were in Suppl. Figs, is now Fig 4 in the main text.

**Also, authors should make clear in MS that "what is original purpose of this MS" in introduction, "which is original findings from this MS" in "Abstract". Original findings of this MS are not clear in the "Abstract"(see below).**

See answer above please, in the abstract section.

**Table 1 showed the result from statistically analyses of the particulate(P) trace metal (TM) data obtained from this study. This statistically analyses are one of the important points for discussion in the MS. However, the detail for the statistically analyses are not clear for reader. For instance, which "day's" data did authors used for evaluating the difference among treatments for each P-TM (d12, d17, d21??)? Which day's data did authors used for evaluating the time difference (is this means significant difference between d12 and d21?)? Please explain more clearly in the text (2.4 section) and in the caption of Table 1. Same for Table 3.**

We used all the days because we performed a Split-Plot ANOVA (or mixed model) which integrates fixed factors (Co2 and Fe) and a repeated measures factor (time) by using the post-hoc Bonferroni, thus, saying that the statistical treatment was a split-plot ANOVA+Bonferroni, compulsory means that time was fully considered during the whole experimental period. We have revised the text though see if this is unclear, and we believe it is not.

- **Abstract**

**Line 30-38; From "Future predicted high…… future ocean dynamics.". This sentence should be supported by results of this study which is described in the one sentence before. Author should state here "what is the original important result and major findings from this study". Present sentence is not clear in this point. I think, main important results and findings from this study, which should be claimed in abstract, are described in line 376-404, and "conclusion". Authors should not include speculation in the "Abstract". The findings should strongly be supported by this study's own results.**

We rewrote the abstract removing most speculations. We tried to be very careful in the abstract as well as in the discussion and conclusions when we talked about particulate metal concentrations (mol L-1) and ratios (Me:P). The abstract has then been modified as follows: *"Rising concentrations of atmospheric carbon dioxide are causing ocean acidification and will influence marine processes and trace metal biogeochemistry. In June 2012, in Raunefjord (Bergen, Norway) we performed a mesocosm experiment, comprised of a fully factorial design of ambient and elevated pCO$_2$ and/or an addition of the siderophore desferrioxamine B (DFB). In addition, the macronutrient concentrations were manipulated to enhance a bloom of the coccolithophore Emiliania huxleyi. We report here the changes in particulate trace metal concentrations during this experiment. Our results show that particulate Ti and Fe were dominated by lithogenic material while particulate Cu, Co, Mn, Zn, Mo and Cd had a strong biogenic component. Furthermore, significant correlations were found between particulate concentrations (mol L$^{-1}$) of Cu, Co, Zn, Cd, Mn, Mo, and P in seawater and phytoplankton biomass (µgC L$^{-1}$), supporting a significant influence of the*

*bloom in the distribution of these particulate elements. The concentrations of these biogenic metals (mol L$^{-1}$) in the E. huxleyi bloom were ranked as: Zn > Cu ≈ Mn > Mo > Co > Cd. Changes in CO$_2$ affected total particulate concentrations (mol L$^{-1}$) and biogenic metal ratios (Me:P) for some metals, while the addition of DFB only affected significantly the concentrations of some particulate metals (mol L$^{-1}$).Variations in CO$_2$ had the most clear, and significant effect on particulate Fe concentrations (mol L$^{-1}$), decreasing its concentration under high CO$_2$. Indeed, high CO$_2$ and/or DFB promoted the dissolution of particulate Fe, and the presence of this siderophore helped maintaining high dissolved Fe. This shift between particulate and dissolved Fe concentrations, in the presence of DFB, promoted a massive bloom of E. huxleyi in the treatments with ambient CO$_2$. Furthermore, high CO$_2$ decreased the Me:P ratios of Co, Zn and Mn, while increased the Cu:P ratios. These findings support theoretical predictions that the Me:P ratios of metals whose seawater dissolved speciation is dominated by free ions (e.g. Co, Zn and Mn) will likely decrease or stay constant under ocean acidification. In contrast, high CO$_2$ is predicated to shift the speciation of dissolved metals associated with carbonates, such as Cu, increasing their bioavailability, and resulting in higher Me:P ratios"*

- **Introduction**

**Line 74-86; I found that some parts of contents in this sentence are overlapped to previous sentence (line 57-71). To make more concise introduction, please compile and reconstruct these two sentences and avoid repetition.**

We have made this more concise and focused and removed a couple of sentences.

**Line 95-105; This sentence is very important for explaining the aim of this MS. In the present description, authors explain "importance of E. huxleyi bloom for biogeochemistry in the ocean". This is the research motivation for study of Segovia et al. (2017), but this is not motivation for investigating particulate trace metals in this MS. Whereas, Authors only describe the aim of this study as "The aim of the present study was …… CO2 and Fe bioavailability." in line 104-105. I think author should add more explanation about "why the measurement of particle trace metals and characterize the change of particle trace metals during E. huxleyi bloom are important?".**

Thanks for this insightful comment. We have clarified the text as follows: *"In the present work a bloom of the coccolithophorid Emiliania huxleyi was induced in a mesocosm experiment in a Norwegian fjord, where the speciation of particulate and dissolved trace metals is very dynamic (e.g. Fe; Ozturk et al. 2002). We aimed to examine and characterize the change of particle trace metals during the E. huxleyi bloom under the interactive effects of increased CO$_2$ and/or dissolved Fe. Emiliania huxleyi is the most cosmopolitan and abundant coccolithophore in the modern ocean (Paasche, 2002) and its growth and physiology has been studied under these experimental conditions (Segovia et al., 2017, Segovia et al., 2018, Lorenzo et al., 2018). Furthermore, E. huxleyi has unique trace metal requirements relative to other abundant phytoplankton taxa (ie. diatoms or dinoflagellates; Ho et al. 2003). Coccolithophores play a key role in the global carbon cycle because they produce photosynthetically organic carbon, as well as particulate inorganic carbon through calcification. These two processes foster the sinking of particulate organic carbon—and trace metals—and contribute to deep ocean carbon export (Hutchings, 2011) and ultimately*

*to organic carbon burial in marine sediments (Archer, 1991, Archer and Maier-Reimer., 1994). However, ocean acidification will disproportionally affect the abundance of coccolithophores, as well as their rates of calcification and organic carbon fixation (Zondervan et al., 2007). The aim of the present study was to characterize the changes in particulate trace metal concentrations—in both lithogenic and biogenic particles—during a bloom of E. huxleyi under realistic changes in $CO_2$ and Fe bioavailability expected by 2100.”*

**Line 97; I could not find "Segovia et al., 2018" and "Lorenzo et al., 2018" in the reference list in end of MS. Please check all reference in the text and reference list in the MS.**

Done

**-        Material and method**

**Line 114, Authors described that CO2 concentration in the mesocosms were measured by NDIR analysis system. Whereas, in line 116-117, authors described that CO2 concentrations in the mesocosms were calculated from pH and alkalinity. Which is correct? According to Segovia et al.2017, "CO2 concentration in inlet flow" is only measured by NDIR.**

Both are correct. The CO2 concentration in the inlet airflow was measured with an IRGA, and the C speciation in seawater was checked  by modelling by using the CO2Calc package from alk and pH according to Segovia at al. 2017.

**Line 121, Why the DFB concentration was set to 70 nM? This value is important for this study because particle dissolution for some trace metal was strongly influenced by the DFB concentrations as authors described in discussion.**

The rational of adding 10-20 times higher DFB concentrations relative to the dFe is very well explained in Marchetti and Maldonado 2016. We have added this reference.

**Line 128, "gentle vacuum pumping". Is this pumping system clean for level of trace metal work? Detailed description is needed for explaining sampling system, cleaning procedure, because these are critical part of trace metal work.**

This has been included as follows in lns217-224: "*All equipment and sampling material used during this study was rigorously acid-washed under trace metal clean conditions and protocols according to GEOTRACES . The material was cleaned with Milli-Q water  (MQw) with 10 % Extran (Fisher Chemicals) at 60°C for 6h, followed by 3 thorough rinses with MilliQ water at room T.  The material was then  cleaned with 10 % HPLC grade HCl (Sigma-Aldrich) at 60°C or 12h and then rinsed thoroughly 5 times with MQw at room T. The material was then covered by plastic and transported to the raft. Sampling in the raft was carried out under a mobile plastic cover hood. Filters were precleaned with 10% trace metal grade hydrochloric acid (Seastar, Fisher Chemicals), at 60°C overnight and were rinsed with MQw”.*

**Line 139, How did authors collect the seawater (by using pumping as described in line 128?).**

Yes

**"0.2 uM" should be change to "0.2um".**

Done

**Line 147; "trace metal hydrochloric acid" should be "trace metal grade hydrochloric acid".**

Done

**Line 154; "0.2 uM" should be change to "0.2um".**
**Done**

**Line 156; Is this "centrifuge tubes" same tube as "2mL PP" tubes?? Reader will be confusing.**

No. They are 2 mL polypropylene tubes. It has been clarified in Ln 232.

**Line 225-249; Section 3.2, 3.3 and 3.4. Authors need to explain detail about statistically analysis for Table 1 and Table 3, see general comment.**

Please see answer above: We used all the days because we performed a Split-Plot ANOVA (or mixed model) which integrates fixed factors (Co2 and Fe) and a repeated measures factor (time) by using the post-hoc Bonferroni, thus, saying that the statistical treatment was a split-plot ANOVA+Bonferroni, compulsory means that time was fully considered during the whole experimental period. We have revised the text though see if this is unclear, and we believe it is not.

- **Discussion**

**Line 286 "well-known metal content in biogenic particles" and line 287 "normalized to Al (mol Me: mol Al in the Earth crust)". Please indicate the number which was used for this calculation and indicate reference for the number.**

We added the appropriate reference in a parenthesis. Also, we believe these are in the legends. We can't add all the ratios in a parenthesis, they too many.

**Line 289; "0.0051 for mol Fe:molP", "0.331 mol Fe : mol Al". How did authors select this number? Which information did author refer? Maybe, "Ho et al., 2006" in Table 2 (A) for "0.0051" and Table 2 (B) for "0.331". Table 2 (B) crustal ratios need reference (Taylor 1964?). Add ref. to "Table 2 (B)".**

The Me:P are those of marine plankton (Ho 2006).We added this to the text.

**Line 321; I could not find "Table 5" in MS.**

Table 5 does not exist anymore, is Table 4. This has been corrected.

**Line 347; "Zn and Cu dissolution (ref.)". Please add reference.**

Added

**Line 353-355; "Rain events…… (Muller et al., 2005)". Do authors have some information on rain event before or during the experiment? If they have, it is important additional information for this interpretation.**

We have added a comment saying that this may be of importance due to the rainy nature of this geographical location

**Line 355-358; ", and indigenous plankton……Knauer et al., 1998).". This sentence is too speculative, and not main discussion for particulate Cu:P ratio. I feel author can delete this sentence.**
Deleted

**Line 362, 363; I can not find "Figure 6" in the MS.**
Fig 6 does not exist anymore. It was a mistake.

**Line 376; "in that it was the only trace element whose particulate concentration significantly and uniquely affected…….". I think this is not true. According to Table 1, other metals (Co, Zn, Cd Mn Mo) were also affected by CO2.**

What we said is true. Particulate Fe (mol L-1) was the only one with a clear CO2 effect, without any interaction with DFB.

**Line 384-387; Authors discussed about partitioning of DFe and PFe with using Figure S2. I think this results and discussion are one of very important finding of this MS. This result is base of predicting future ocean in the following discussion. I recommend that Figure S2 should be placed for the normal Figures (not in Supplementary).**

We thank reviewer for this suggestion. We have included the figure in the main text now.

**Line 396; I could not find Table 5 in the MS.**
Table 5 does not exist anymore. It was a mistake; it is Table 4.

**Line 424; "due to increased Fe bioavailability". How authors judged the increasing Fe bioavailability in the LC+DFB treatment? Is there any evidence which indicate that DFB-bounded Fe is available for E. huxleyi? Please discussed more carefully about the changing Fe bioavailability in "discussion".**

We removed the bioavailability and avoided any speculation.
* * *
We thank the reviewers for their constructive comments and their time, and we hope that our responses are satisfactory

Yours sincerely,

Maria Segovia and Maite Maldonado

[revised manuscript text omitted]

---

## Author Response (AR3)

**RESPONSES TO REVIEWERS COMMENTS FOR BG-2018-448 by Lorenzo et al. 3ʳᵈ resubmission**
* * *
Referee #1

**Review comments for "Particulate trace metal dynamics in response to increased CO2 and iron availability in a coastal mesocosm experiment".**

**The authors have improved this manuscript with addressing my comments. My serious concern is mostly resolved. However, the authors did not resolve all of my concerns. I still find several points which need to be improved before publication.**

**I have read this version of manuscript carefully, and I still feel that explanation of statistically analysis for Table 1 is not kind for reader. Why author do not describe following information in the text or caption of Table 1? "We used all the days because we performed a Split-Plot ANOVA (or mixed model) which integrates fixed factors (Co2 and Fe) and a repeated measures factor (time) by using the posthoc Bonferroni, saying that the statistical treatment was a split-plot ANOVA+Bonferroni, compulsory means that time was fully considered during the whole experimental period."**

**They have made the response only to reviewer's comments. This information is very important for reader's understanding that how the statistically analysis have done.**

Thanks for the comment. It has been included now as follows:
*"Table 1. Statistical analyses (Split-plot ANOVA) of the effects of high $CO_2$, the addition of DFB, and their interaction, as well as the effect of time, on the concentrations of particulate metals (mmol $L^{-1}$, data in Table S2, and Figure 3) in particles collected from the different mesocosms treatments. We used all the days for the analyses because the Split-Plot ANOVA integrates fixed factors (CO2 and Fe) and a repeated measures factor (time) by using the post-hoc Bonferroni, thus, time was fully considered during the whole experimental period."*

**Line 125 "By day 17,......and/or the addition of DFB (Chen et al., 2004)." Increased dissolved Fe by adding DFB is bioavailable?**

DFB enhances Fe solubility (Chen et al. 2004), which increases the dFe pool. Given the positive response of E. huxleyi growth in the low CO2, high DFB treatments, the DFB seems to have improved the bioavailability of Fe to E. huxleyi. This is supported by previous studies showing that *E. huxleyi* produces a wide range of compunds with high affinity for Fe (Boye & Van den Berg 2000). Furthermore, *E. huxleyi* is able to acquire Fe from a variety of organic Fe complexes (Hartnett et al. 2012), including Fe-DFB (Shaked & Lis 2012, Lis et al. 2015).

This has been clarified in the first paragraph of the discussion, although it was also amended around Ln 125.

**Is DFB-Fe can be detected by CL-FIA which described in Segovia et al., 2017? Strong chelate like DFB prevent dissolved Fe detection measuring by resin preconcentrate-CL-FIA measurement system. It should be made clear that which chemical species do**

**authors describe as for dissolved Fe in this study (Is DFB-bounded Fe included in this dissolved fraction, or not?). Also, it is necessary to clearly describe how do authors think about that how the availability of iron was changed by adding DFB.**

We have included the following paragraph for clarification in the text, Ln 147-153:

*"The pH of the 0.2 μm filtered DFe samples was lowered to 1.7 (using SeaStar® HCl) upon collection. Lowering the pH to 1.7, with HCl for more than 24 hours, ensures solubilisation of all the Fe in the sample, as well as the release of all the Fe bound within strong organic complexes (such as Fe-DFB), thus making all DFe available for analysis (Johnson et al. 2007). During flow-injection analysis with chemiluminescence detection (FIA-CL), the sample is only buffered to a higher pH immediately before entering the flow cell, right in front of the photomultiplier; so that Fe-DFB complexing kinetics are sufficiently slow to allow total DFe to be measured"*

Johnson et al. 2007. Developing Standards for Dissolved Iron in Seawater. EOS Transactions American Geophysical Union 88 (11): p. 131-132

**Some previous studies indicate that DFB-bounded Fe is not available, as authors described in the text. Is DFB-Fe available for E. Huxley? If so, please indicate a reference. Or, do adding DFB induce other chemical species of dissolved fraction? Author should describe this aspect clear, because this point is very important for this study.**

Reviewer is right. This is a crucial point. The following clarification has been inserted in the text in Lns 264-268:

*"Our results suggest that E. huxleyi is able to utilise DFB-bound Fe (Fe-DFB) due to the dynamics observed in Segovia et al. 2017. Indeed, E. huxleyi has been shown to produce a wide range of metabolites which are organic complexes with high affinity for Fe (Boye & Van den Berg 2000), and, E. huxleyi is also able to acquire Fe from organic Fe complexes (Hartnett et al. 2012) including Fe-DFB (Shaked & Lis 2012, Lis et al. 2015)"*

**Line 130-132, "Water samples from......onshore laboratory." This sentence should be moved in section 2.3.1.**

We are not taking this suggestion because section 2.3.1 only refers to pMe sampling. The paragraph reviewer refers to concern to all variable samplings, not only pMe.

**Authors should indicate manufacture and model information for the "vacuum pump".**

Included in Ln 132:

*"…pumping of 25 L volume into acid-washed carboys by using membrane vacuum pumps (PALL) working at reverse flow. Carboys were quickly transported…"*

**Line 145, "for this very experiment.". What is "very".**

"Very" means "the same". Grammar is correct.

**Line 155, Authors should indicate the type of "Filters". Is this also AcroPac Supore, but membrane type?? Size??**

It has been specified as follows:

"*Seawater was collected from each mesocosm, filtered through AcroPak® capsule filters with  0.2 μm Supor® membrane into the trace metal clean LDPE bottles*".

**Line 158, "without manipulation" should be changed to "without oxalate-EDTA wash".**
**Line 164, Authors should indicate material and volume of "centrifuge tubes"**

Added:

"*...directly to 2 mL centrifuge polypropylene tubes for storage…*"

**Line 178, blank value should be appeared in the Supplemental material such as S-Table 1.**

As indicated in the Materials and Methods "Filter blanks were collected and subjected to the same storage, digestion, dilution, and analysis processes, and these blank values were subtracted from sample measurements" These filters blanks were collected every time we collected samples, and the samples collected in a given day were corrected with the value of the blanks collected that same day. Thus, it is practically impossible for us to include the blanks for all these measurements in Supplementary Table 1. They were inherently included!

While writing our manuscript, we also searched similar particulate metal manuscripts and we reported the data as reported in those manuscripts. The blanks are not reported in the data tables, as the data are corrected already for them (e.g. Cid et al. 2012, *J Oceanogr*. 68:985–1001; Ohnemus and Lam 2012, *Deep Sea Research Part II: Topical Studies in Oceanography* 116: 283-302; Ho et al. 2007, *Limnol. Oceanogr*., 52(5): 1776–1788).

In light of this, we think our data are correct in the present form.

**Line 215-216, "This diatom bloom was associated with a sharp decrease in nitrate and silicate acid concentration". I think this is not correct. Why nitrate decrease with diatom decreasing? Silicate have not sharp decreased during diatom decreasing.**

Silicate was first consumed. When we measured day 0  and day 1, silicate was already gone. Then nitrate was consumed. Iron requirements of phytoplankton are strongly influenced by the availability and source of nitrogen (Maldonado and Price 1996, Schoffmann et al. 2016). Phytoplankton that is utilizing nitrate ($NO_3^-$) has higher Fe requirements than phytoplankton utilizing ammonium ($NH_4^+$) for growth (Maldonado and Price 1996, Schoffmann et al. 2016). $NH_4^+$ can be directly incorporated into amino acids, while extra iron is needed for nitrate assimilation, because nitrate and nitrite reductase contain Fe cofactors. In addition, the energy for $NO_3$ reduction is produced by the Fe-rich photosynthetic electron transport chain. Thus $NO_3^-$ was consumed by other phytoplankton groups not limited by Fe harming diatoms that were already silicate limited.

This is clearly explained in Segovia et al.,2017 so we refer to our paper for further details.

**Line 327, "Figure 4" should be changed "Figure 3".**

Done

**429-430, "promoted a massive bloom of E huxley in the treatment with ambient CO2, due to increased dissolved Fe". It should be made clear that which chemical species do authors describe as for dissolved Fe in this study (Is DFB-bounded Fe included in this dissolved fraction, or not?). Some previous studies indicate that DFB-bounded Fe is not available, as authors described in the text. Is DFB-Fe available for E. Huxley? If so, please indicate reference. See comment above too.**

This has been answered before.

**Line 435-436, "The decrease in particulate Fe may..... in open ocean setting.". Delete this sentence. This is not a conclusion from this study. No data from this study indicate this.**

Done

**End of review.**
* * *
We thank the reviewers for their constructive comments and their time, and we hope that our responses are satisfactory

Yours sincerely,

Maria Segovia and Maite Maldonado

[revised manuscript text omitted]

[revised manuscript text omitted]

---

## Author Response (AR4)

**RESPONSES TO REVIEWERS COMMENTS FOR BG-2018-448 by Lorenzo et al. 4rd resubmission**

Dear Editor,

We have amended all the small issues that were indicated. We hope that the manuscript is now satisfactory for publication

Yours sincerely,

Maria Segovia and Maite Maldonado